# The Knowledge and Application of Sedimentary Conditions of Shallow Marine and Tidal Waters of Ionian Islands, Greece: Implications for Therapeutic Use

Chrysanthos Botziolis [1,*], Nicolina Bourli [1], Elena Zoumpouli [1], Penelope Papadopoulou [2], Nikolaos Dimopoulos [1], Andriana Kovani [1], Panagiotis Zelilidis [1], Diamantina Christina Aspioti [3], George Iliopoulos [2] and Avraam Zelilidis [1]

[1] Laboratory of Sedimentology, Department of Geology, University of Patras, 26504 Rion, Greece; geo12034@ac.upatras.gr (N.D.); a.kovani@ac.upatras.gr (A.K.); a.zelilidis@upatras.gr (A.Z.)
[2] Laboratory of Paleontology and Stratigraphy, Department of Geology, University of Patras, 26504 Rion, Greece; penelpapadop@upatras.gr (P.P.); iliopoulosg@upatras.gr (G.I.)
[3] Laboratory of Hydrocarbons Chemistry and Technology, Department of Mineral Resources Engineering, Technical University of Crete, 73100 Chania, Greece; daspioti@tuc.gr
* Correspondence: cbotziolis@ac.upatras.gr

**Abstract:** This study delves into the sedimentation mechanisms governing mud deposits in shallow marine and tidal environments, with a particular focus on elucidating the versatile therapeutic applications of these muds. This research provides valuable insights for optimizing the selection of mud as a cosmetic resource that can positively influence human health and well-being by utilizing a comprehensive analysis involving $CaCO_3$, TOC, grain size, and statistical parameters across six outcrops situated on the Kefalonia and Corfu islands. The research reveals that the $CaCO_3$ content of mud deposits on both islands is comparable. Despite the average value (26.71%) significantly exceeding the recommended value (10%) for optimal plasticity, no discernible impact on the mechanical behavior and plasticity of the clay was observed, rendering it a neutral quality criterion. Notably, the TOC content is higher on Corfu Island, suggesting its potential superiority for mud therapy. However, all samples exhibit a TOC content (<0.77%) considerably below the threshold required (2–5%) for material maturation in mud therapy. Consequently, an enrichment of samples with organic matter is imperative. The application of statistical parameters, analyzed through graphical methods, facilitated the creation of various bivariate diagrams, offering insights into the prevailing environmental conditions during deposition. Linear and multigroup discriminant analyses categorize two sediment types: a unimodal type, characterized by mud grain-size dominance, deposited in a shallow water environment, and a bi-modal type, featuring mud and sand content, deposited in a tidal-affected environment. This classification underscores the potential of shallow marine muds (Kefalonia Island) for therapeutic use, given their optimal grain size. In contrast, the tidal mud (Corfu Island), while also suitable for mud therapy, necessitates processing as a cosmetic product to minimize sand content, as coarser fractions may induce skin irritations or injuries.

**Keywords:** mud therapy; calcium carbonate; total organic carbon; grain-size analysis; statistical parameters; Ionian Islands; Kefalonia Geopark; Greece





## 1. Introduction

The sedimentation of mud in shallow waters, especially in tidal environments, is a subject of significant interest to the scientific community [1]. Coastal and estuarine sediments mainly consist of fine-grained sediments collectively known as mud, which includes silts and clays. These dynamic systems are characterized by the regular ebb and flow of tidal waters and serve as complex interfaces where terrestrial and marine processes intersect. The deposition and resuspension of mud in tidal environments are a result of the

dynamic interplay of various factors, such as tidal hydrodynamics, sediment supply, and the influence of benthic organisms [2–4].

Apart from contributing to the geological composition of tidal environments, these fine-grained sediments also possess unique therapeutic properties that have been recognized and utilized for centuries [5–8]. The documented use of mud in spa treatments and medical therapies includes applications like mud baths, wraps, and localized treatments for dermatological and musculoskeletal ailments [9]. These therapeutic uses are rooted in the distinct mineralogical, chemical, and physical characteristics of coastal mud deposits, which may contain elevated levels of elements such as sulfur, selenium, iodine, and organic matter [10]. Scientific investigations have shown that these properties offer various health benefits, including anti-inflammatory, pain relief, and detoxification effects.

The study of coastal geology and mud sedimentation in shallow tidal waters is of great significance to various professionals, including geologists, ecologists, and professionals in the medical and wellness fields. The unique therapeutic attributes of mud from these ecosystems open new opportunities for interdisciplinary research. Recognizing the multifaceted importance of mud sedimentation in shallow tidal waters is essential, as it underscores the potential of therapeutic mud as a resource that can positively influence human health and well-being. An integrated approach to the study of these environments is necessary, as it emphasizes the multifaceted importance of these systems [1,2,9,10].

Several studies have investigated the depositional processes of mud in coastal settings. These studies have shown that the transport and deposition of mud are controlled by various factors such as the supply of sediment, hydrodynamic conditions, and biological activity [2–4]. In addition, the physical and chemical properties of the sediments, such as grain size distribution, calcium carbonate and organic content, play a significant role in the depositional processes of mud [11–13]. Understanding these factors is crucial in predicting the spatial and temporal distribution of mud in coastal settings.

The scientific significance of sedimentary environments in relation to therapeutic mud lies in the intricate composition and properties of mud formed within these settings. Thus, gaining a comprehensive understanding of the sedimentary processes and the characteristic of mud is essential to understand the potential of therapeutic mud as a resource that can positively influence human health and well-being [14–16]. Investigating the sources, transport, and depositional processes of mud in coastal settings is crucial, as it underscores the multifaceted importance of mud sedimentation in shallow tidal waters. An integrated approach to the study of these environments is necessary, as it emphasizes the multifaceted importance of these systems [1,2,9–13]. This study focuses on the properties of mud along the coasts of Xi and Koutala (Kefalonia Island) and Arillas and Ag. Stefanos (Corfu Island) areas, that are commonly utilized for mud therapy by the local population and tourists. The primary goal is to pinpoint equivalent areas suitable for exploitation while ensuring the preservation of these popular coastal attractions. While the above-mentioned characteristics are significant in mud therapy exploration, a comprehensive approach involves additional analysis, such as mineralogical features, particularly the clay minerals, as well as major, trace, and rare earth elements analysis, pH, plasticity, surface characteristics, and thermal properties. Due to the extensive database obtained, a detailed account is presented in [17], which will offer a comprehensive overview of the mineralogical, geochemical, and physical characteristics of the materials, providing a detailed perspective on their suitability for therapeutic applications.

## 2. Geological Setting

Western Greece has a complex geotectonic evolution linked to the different types and movements of the involved tectonic plates. The Pindos orogeny (Figure 1), a thrust system, serves as the boundary between the external and internal Hellenides, resulting from the collision of the Apulia and Pelagonia plates following the closure of the Pindos Ocean [18–22]. The separation of the Apulia plate from Gondwana involved a system of normal faults-oriented northeastward, offset by transform faults in a northwest direction, leading to the

creation of the Neo-Tethyan Ocean during the Jurassic period [23,24]. During the Upper Jurassic, a series of platforms and basins developed, as evidenced by the spread of pelagic and deep-sea carbonate formations (Figure 2), including slope deposits, e.g., [21,25–31].

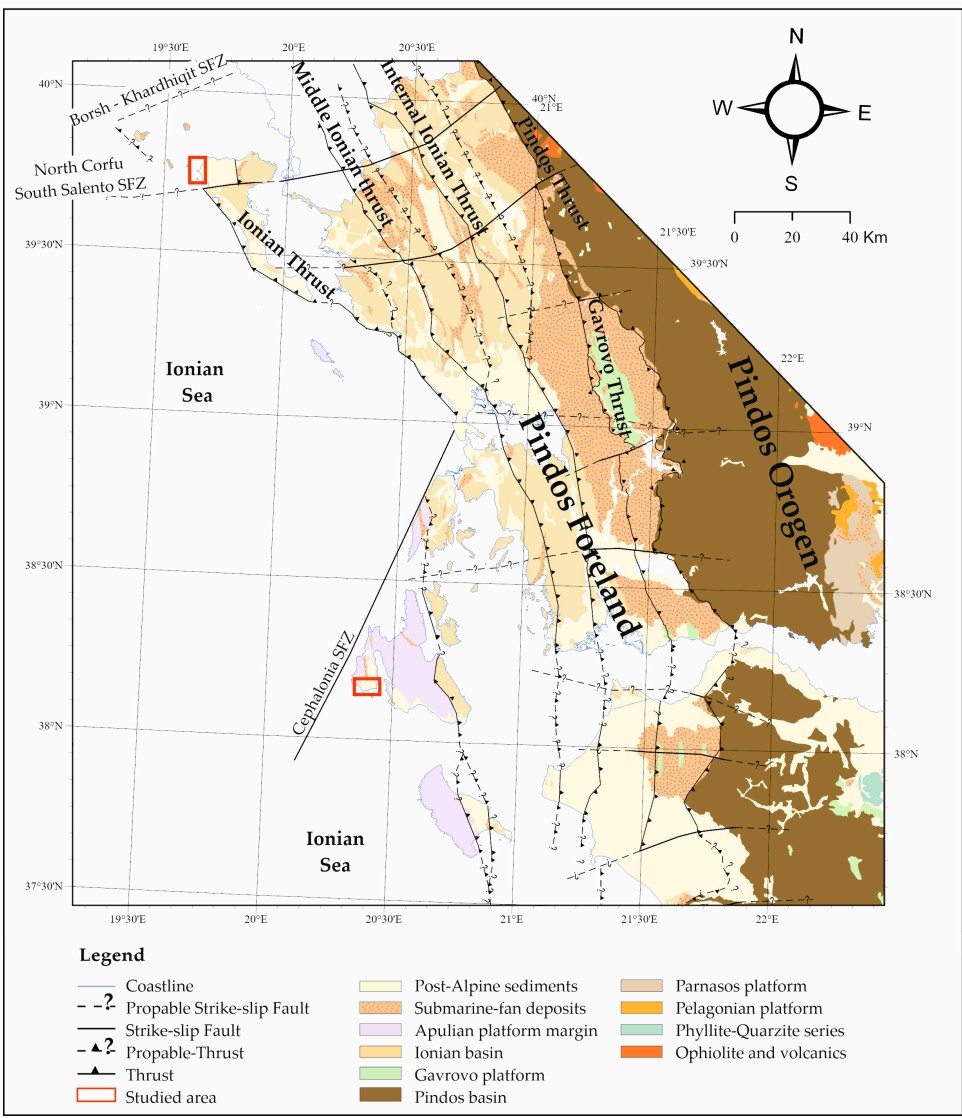

**Figure 1.** Geological map of Western Greece (modified from [22]), illustrating the major structural elements. The red frames show the studied areas.

The Ionian Basin (IB) developed between the Pre-Apulian (or Apulian Platform Margin—APM) and Gavrovo platforms. It is covered by a substantial sediment thickness that extends over a significant portion of western Greece (Figures 1 and 2a). Stratigraphic examination of the IB (Figure 2a), reveals its progression through distinct evolutionary stages [21,22,28,30–33]. A pre-rift stage dating back to pre-tectonic movements, involved in evaporites from the Lower to Middle Triassic and shallow-water limestones from the Upper Triassic to Lower Jurassic period. During this phase, the IB represented a shallow peri-tidal environment, periodically experiencing emergence [34]. Up until the Lower Jurassic, the IB constituted the western part of an extensive carbonate continental platform that extended across the entirety of western Greece. Notably, the eastern boundary of this continental platform corresponds to the Gavrovo platform, given the existence of the Pindos Basin since the Triassic [21,22,28,30–33].

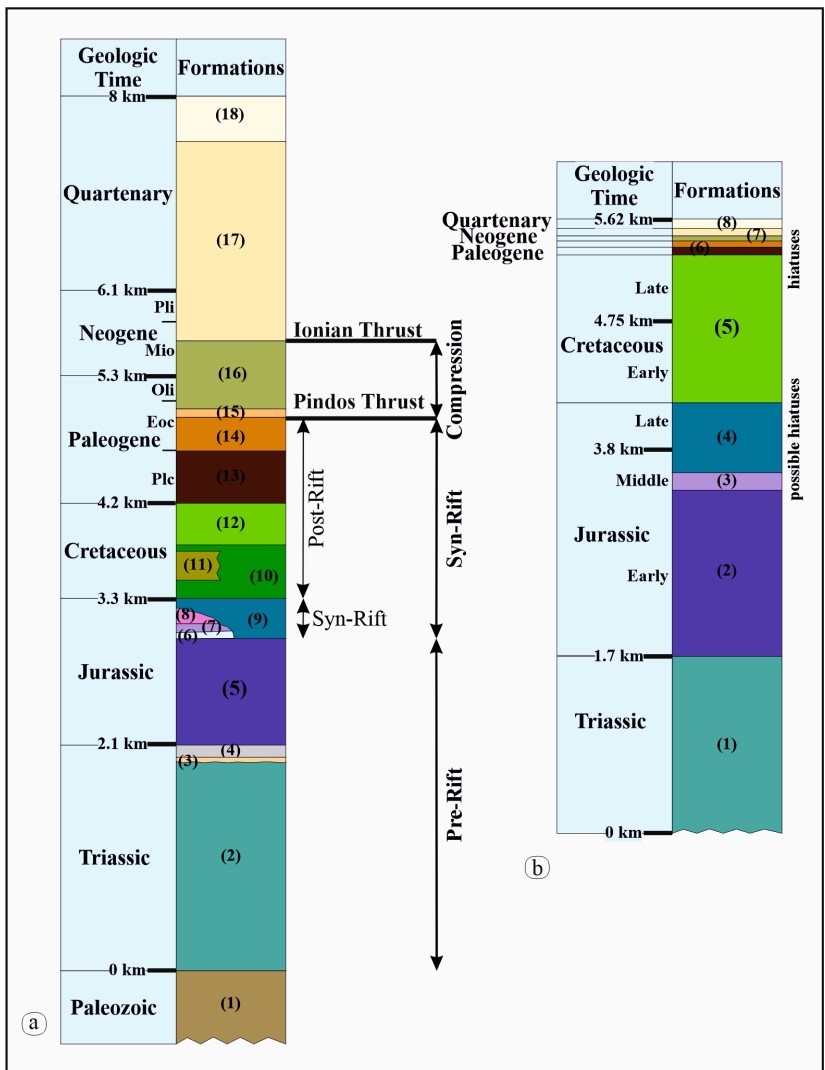

**Figure 2.** Lithostratigraphic column of (**a**) the IB (Formations: (1) continental clastic, (2) evaporites, (3) breccia, (4) "Foustapidima" limestone, (5) "Pantokrator" limestone, (6) "Ammonitico Rosso", (7) Limestone with filaments, (8) Posidonia beds, (9) "Louros and Siniais" limestone, (10) "Vigla" limestone, (11) "Vigla" shale, (12) Senonian limestone with microbreccia and calciturbidites, (13) limestone with microbreccia and calciturbidites, (14) platy limestone, (15) deep-sea fans deposits, (16) deltaic deposits, (17) alluvial and recent deposits) (modified after [30,31,33]), and (**b**) the APM (Formations: (1) evaporites, dolomites, shale intercalations, (2) limestones, dolomitic limestones, anhydrites, shales intercalations, (3) thin bedded limestones with cherts, marly limestones, marls, anhydrites, (4) limestones, marly limestones, marls, (5) undifferentiated limestones with cherts, brecciated limestones, (6) pelagic limestones with breccia, (7) marls, marine marls and sands, (8) marine marls) (modified after [35]).

This stage is followed by a subsequent syn-rift phase primarily characterized by the presence of limestones from the Lower Jurassic to the Lower Cretaceous (with occasional layers of marl), showing a general subsiding trend. The tectonic activity during the syn-rift phase resulted in the formation of half-grabens, leading to notable variations in the thickness of deposits. Throughout the Upper Cretaceous, the Ionian Basin (IB) experienced substantial tectonic activity. This activity gave rise to asymmetric troughs due to both compressional and extensional faults. The raised blocks resulting from this tectonic activity acted as significant sources of sediment within the basin. Consequently, the basin received a substantial influx of coarse clastic sediments. During the Upper Cretaceous, the raised blocks in the asymmetric troughs displayed increased activity, leading to the erosion

of deposits from the "Vigla" formation of the Lower Cretaceous and the formation of (micro-) limestone turbidite deposits [30,31]. The thickness of these (micro-) limestone and limestone turbidite deposits varied because of the basin asymmetry and fluctuations in water depth [30,31]. Extensional tectonic activity played a crucial role in the evolution of the basin and is believed to have been active in the IB from the Upper Jurassic to the Lower Eocene [30,31].

The shift from extension to compression in Western Greece occurred during the Middle Eocene [33], leading to the formation of the Pindos Foreland Basin (PFB) (Figure 1). This transformation was caused by alterations in plate configurations, including the closure of the Mesozoic Neotethyan Ocean and the convergence of the Apulian and Eurasian plates [18,20]. The geodynamic environment of the external Hellenides shifted from being a continental rift to becoming a fold and thrust belt (FTB), accompanied by the establishment of a foreland basin system, as discussed in previous studies [20–22,28,32,33,36–39]. Within this context, the Pindos Orogen represents the FTB, while the western region, previously identified as the IB, now corresponds to the PFB, as documented in Botziolis et al. [36,37]. In this configuration, transitional beds of calcilutite and calcarenite from the Lower to Middle Eocene pass up-section to submarine fans, with an approximate thickness of 2000 m [36,37]. Compression occurred in the western section of the external Hellenides from the Middle Eocene to the Early Miocene, while the eastern section continued to experience extensional tectonics [40–44].

West of the IB, the APM positioned between the Apulian Platform (AP) and the IB marks the transition to the AP [35], which is also present in Italy. The APM is characterized by a continuous sequence of neritic carbonate rocks (Figure 2b), with deposition commencing in the Upper Triassic and extending through the Oligocene. Surface exposures of these formations are of Upper Cretaceous age, while older formations have only been observed through onshore or offshore drilling. The Upper Cretaceous to Oligocene carbonate sequence is defined by limestone turbidite deposits of lower-downslope facies and (micro-) limestone horizons alternating with pelagic limestones, attributed to paleogeographic positioning in the lower-downslope area of the AP. A distinctive feature is the absence of submarine fan deposits, which were replaced by a series of marly formations interbedded with (micro-) limestone calcite. In summary, the APM, during its evolution from the Triassic to the Eocene, received neritic sediments without undergoing significant changes in its character.

## 3. Materials and Methods

The research area encompasses both coastal and inland regions located on the southern Paliki Peninsula of Kefalonia Island and the western-northwestern shoreline of Corfu Island. We conducted an in-depth analysis of sedimentological characteristics, supplemented by illustrative outcrop photographs, focusing on six synthetic outcrops. These outcrops vary in length (5 to 300 m) and height (4 to 25 m). Some show lateral continuity, extending over hundreds of meters and being interconnected or closely spaced with distances of less than 10 m, while others are more widely separated. Lithology, detailed bed thickness, grain size and sedimentary structures were graphically recorded in sedimentary logs.

Over 200 mudstone samples from all studied sections and 5 sand samples from Xi beach sand were collected. For micropaleontological analysis, 60 samples were selected from various sections (20 samples from the Xi section, 2 samples from Chavdata, 2 samples from Matzavinata, 3 samples from Koutala-Kefalonia Island, 10 samples from Arillas north, 13 samples from Arillas South, and 17 samples from Agios Stefanos). Sediment underwent wet sieving through 64 μm and 500 μm mesh sieves with tap water and was subsequently oven dried on 40 °C. Foraminifera analysis involved an Optica LAB20 stereoscope for taxonomic identification based on established criteria according to [45,46] (references therein). The categorization of species was performed based on their ecological preferences, drawing from the criteria established by [47,48] and referenced works like [49–53] and references therein.

Geochemical analysis of 120 samples (42 samples of mud and 1 sample of sand from the Xi section, 15 samples from Chavdata, 15 samples from Matzavinata, 7 samples from Koutala-Kefalonia Island, 5 samples from Arillas north, 15 samples from Arillas South, and 20 samples from Agios Stefanos) included determining calcium carbonate ($CaCO_3$) and total organic carbon (TOC) content. $CaCO_3$ analysis utilized $CH_3COOH$ decomposition, according to Varnavas [54] method, while TOC analysis employed Rock-Eval II and VI (Delsi Inc., Edison, NJ, USA) analyzers under standard conditions, by utilizing ~100 mg of pulverized rock. The samples were then heated in a helium atmosphere, using a suitable oven [55–57]. Additional analysis, including mineralogical features, as well as major, trace, and rare earth elements analysis, pH, plasticity, surface characteristics, and thermal properties were also conducted and detailed methodology is outlined in [17].

Grain-size analysis of 125 samples (43 samples of mud and 5 samples of sand from the Xi section, 15 samples from Chavdata, 15 samples from Matzavinata, 7 samples from Koutala-Kefalonia Island, 5 samples from Arillas north, 15 samples from Arillas South, and 20 samples from Agios Stefanos) was conducted for particles larger and smaller than 63 mm using sieve and pipette analysis methods, respectively. Statistical parameters such as mean, sorting, kurtosis, and skewness were computed using Origin Software, and cumulative gradation curves were generated with Grapher software v.16.2.354.

Sediment lithological characteristics were categorized according to the Folk and Ward classification [58]. Statistical analysis, including Linear Discriminant Analysis (LDA) proposed by Sahu [59], was employed to interpret changes in energy and fluidity during or before sedimentation, revealing correlations with geological processes and depositional environments.

To discriminate between shallow turbulent waters (SA) and beach (B) environments, Equation (1) was applied:

$$Y1(SA:B) = -3.5688 \times M_d + 3.7016 \times \sigma_1{}^2 - 2.0766 \times SK_1 + 3.1135 \times K_G, \tag{1}$$

If the value of Y1 is less than $-2.7411$, the environment is classified as "shallow turbulent waters." Conversely, if Y1 is greater than $-2.7411$, it is categorized as a "beach" environment.

For distinguishing between beach (B) and shallow sea (SM), Equation (2) was employed:

$$Y2(B:SM) = 15.6534 \times M_d + 65.7091 \times \sigma_1{}^2 + 18.1071 \times SK_1 + 18.5043 \times K_G, \tag{2}$$

When the value of Y2 falls below $-63.3650$, it indicates a "beach" environment, while a value above $-63.3650$ suggests a "shallow sea" environment.

To discriminate between shallow sea (SM) and deltaic or lacustrine (L) depositional environments, Equation (3) was utilized:

$$Y3(SM:L) = 0.2852 \times M_d - 8.7604 \times \sigma_1{}^2 - 4.8932 \times SK_1 + 0.0482 \times K_G, \tag{3}$$

If the value of Y3 is greater than $-7.4190$, the environment is categorized as "shallow sea." In contrast, if Y3 is less than $-7.4190$, it is identified as a "deltaic or lacustrine" environment.

To distinguish between deltaic (D) and turbid deposits (T), Equation (4) was applied:

$$Y4(D:T) = 0.7215 \times M_d - 0.4030 \times \sigma_1{}^2 + 6.7322 \times SK_1 + 5.2927 \times K_G, \tag{4}$$

If the value of Y4 is less than 9.8433, it indicates a "turbid deposit" environment, while a value greater than 9.8433 suggests a "deltaic deposit" environment.

## 4. Sedimentary Environments

### 4.1. Kefalonia Island

In the southern region of the Paliki peninsula (Figure 3), three distinct sedimentary cycles from the Pliocene period can be identified [60,61]. The lower cycle primarily com-

prised limestone and gypsum, followed by the middle cycle composed of blue marls, clays, and marly limestone, occasionally interbedded with sand and the upper cycle consisting of limestone and sand. These formations lay conformably to the Miocene limestone, and their boundary is marked by crystalline gypsum bodies closely associated with marly marls.

Within the coastal zone spanning from Koutalas to Xi Beach (Figures 3b and 4a,b), the middle cycle is found, featuring marls alteration with clay and thin sandstone (Figure 5), posing a cumulative thickness of 300 m. Near the base (Koutalas Beach) (Figure 5Aa), a limestone bed is observed, transitioning upwards to yellowish sandstone, and marly limestone, occasionally interbedded with thin marls (Figure 5Ab), rarely containing fossil-rich beds (Figure 5Ac). Up-section, predominantly at Xi Beach (Figure 5Ba), there are cyan-colored clays rarely interbedded with sandstone. These beds often display lenticular lamination (Figure 5Bb), with sandstone beds less than 10 cm thick. Sporadic, muddy-rich debris flow occurs (Figure 5Bc), on top of fossil-rich layers (Figure 5Bd) and eroding the top of parallel laminated sandstone (Figure 5Be). In the Matzavinata section (Figures 3b, 4c and 6A), there are alternating layers of clays and marls (Figure 6A), rarely interbedded with yellow limestone on the top of the section. At the top of the sequence, the Chavdata section (Figures 3b, 4d and 6B) exhibits dark to light gray clays passing up-section to alternations of gray, blue, green, and red clays, and to yellowish sandy clays (Figure 6Ba). These beds are often bioturbated, containing vertical burrows (Figure 6Bb) and *Thallasinoides* isp. (Figure 6Bc).

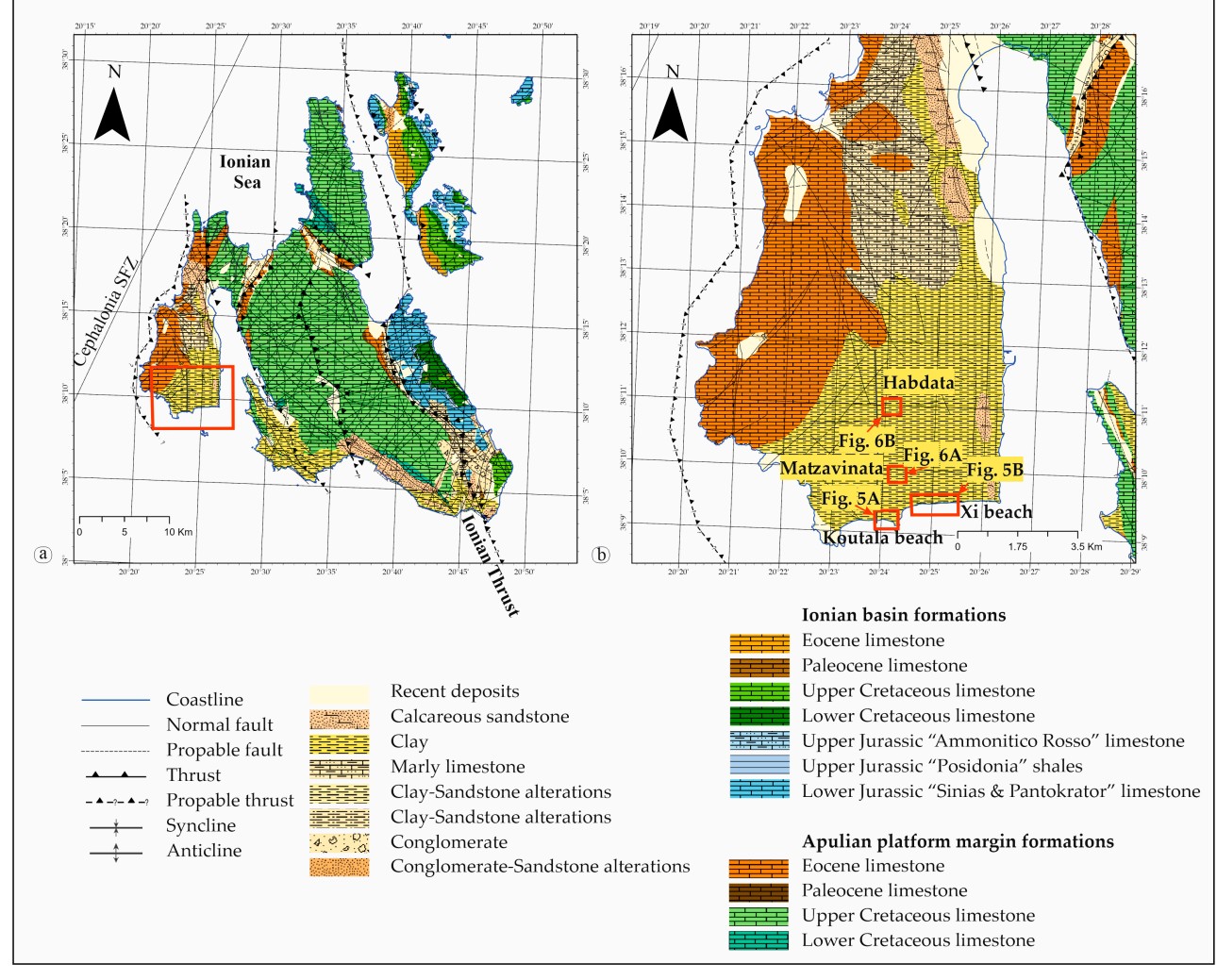

**Figure 3.** (**a**,**b**) Geological map of Kefalonia Island (modified after [31]). The red frame mark the studied areas.

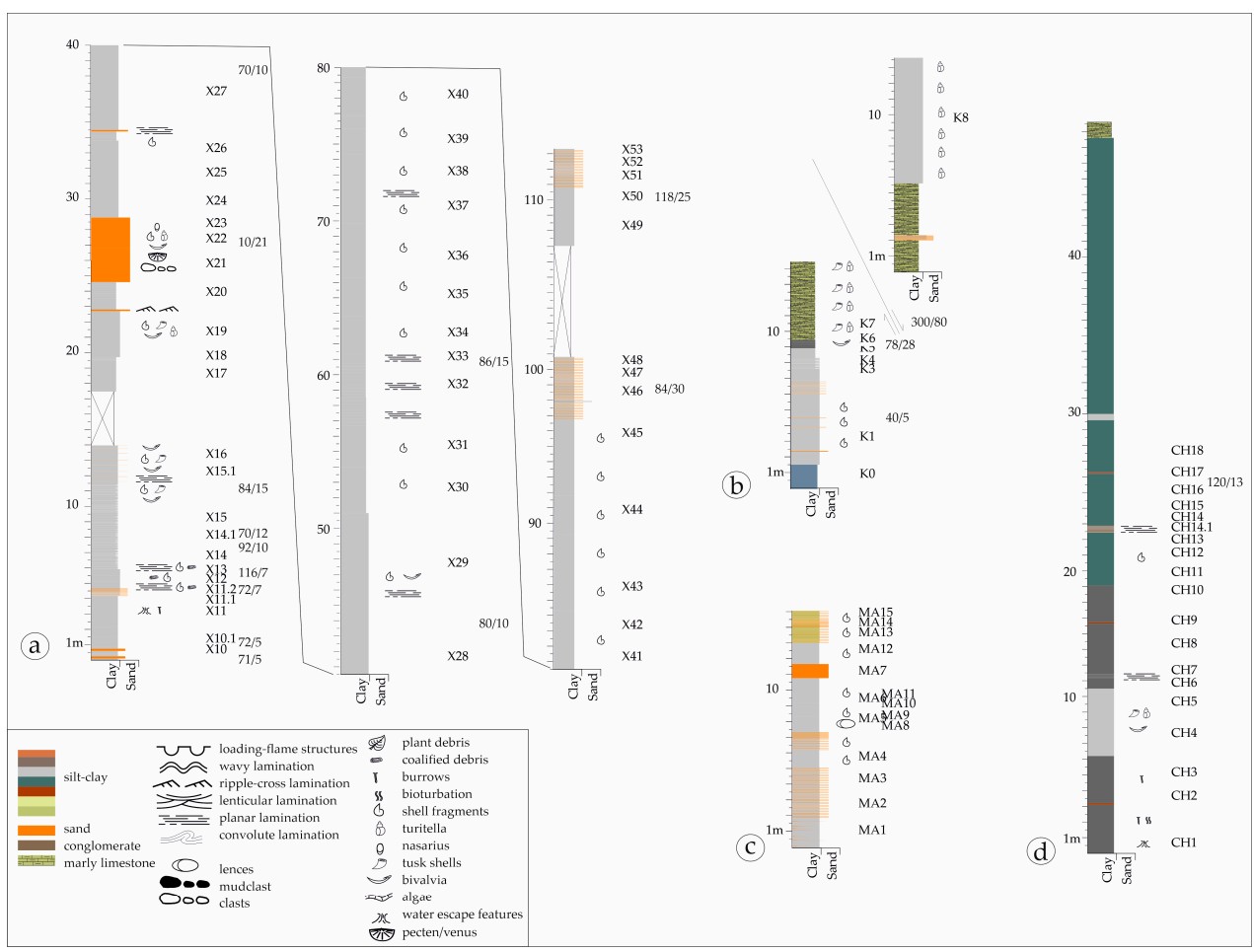

**Figure 4.** Representative stratigraphic logs (Supplementary Materials Figures S1–S4) of the (**a**) Xi beach, (**b**) Koutala beach, (**c**) Matzavinata and (**d**) Chavdata studied sections.

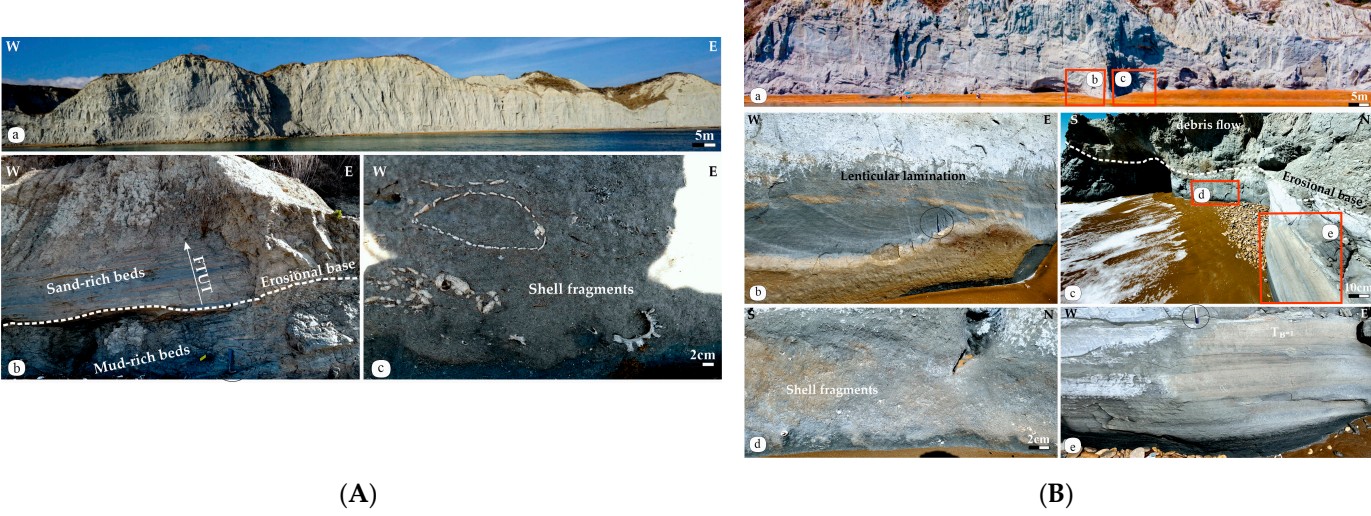

(**A**)                                                                 (**B**)

**Figure 5.** (**A**) (**a**) Panoramic view of Koutala beach section; (**b**) Sand-rich beds with erosional beds and characteristic Finning and Thinnin upward trend, abbreviations FTUT: Finning and Thinnin upward trend; (**c**) fossil-rich muddy beds.; (**B**) (**a**) panoramic view of Xi beach, depicting Figure b, c locations; (**b**) lenticular laminated sandstone; (**c**) muddy debris flow; (**d**) sandstone with shell fragments; (**e**) parallel laminated sandstone.

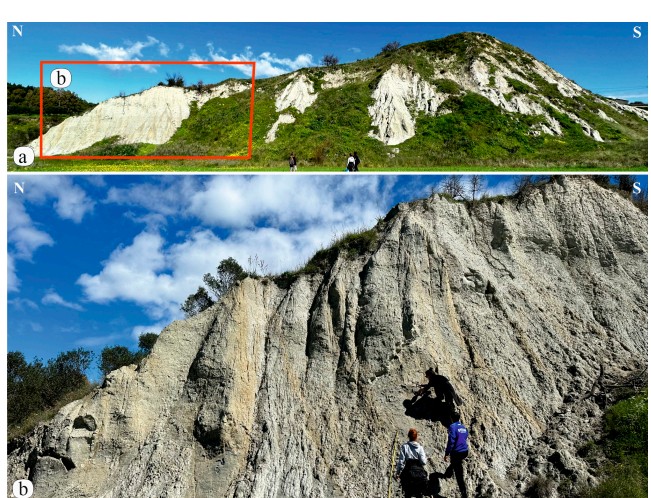

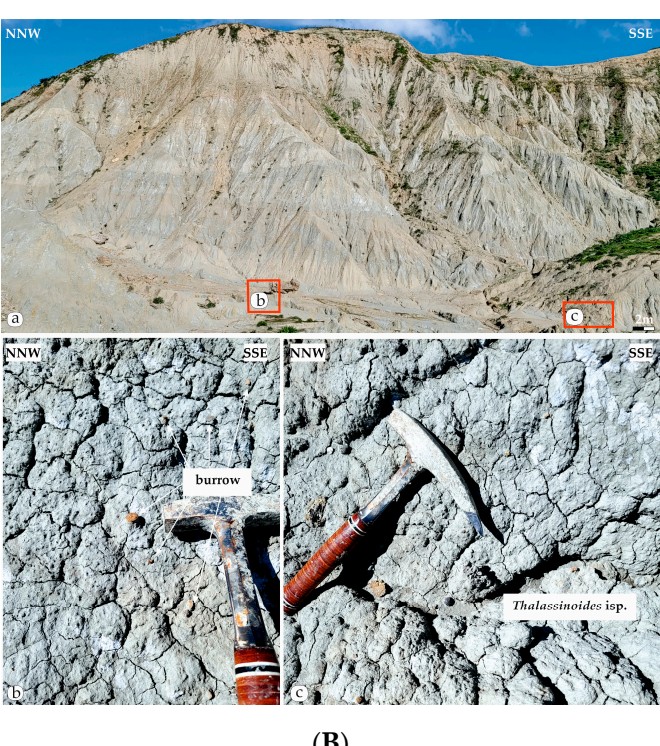

(**A**)　　　　　　　　　　　　　　　　　　　　(**B**)

**Figure 6.** (**A**) (**a**,**b**) Panoramic view of Matzavinata section.; (**B**) (**a**) panoramic view of Chavdata section, depicting Figure b, c locations; (**b**) burrows in mudstone; (**c**) *Thallasinoides* isp.

At Lixouri area, the deposits reach a thickness of 800 m and consist of clays, marls, and marly limestone, occasionally interbedded with sandstones. South of Lixouri, marls are characterized by sparse limestone interbedded with sandstone, reaching a total thickness of approximately 300 m. These deposits are interpreted as transgressive lacustrine or marine sediments, following an uplift and erosion of the Miocene [60,61]. More specifically, the marine phase is evident in the west and southwest of the island, featuring marls and sandstones with a thickness of 300 m, followed by a series of Pleistocene deposits with a thickness of 600 m. According to Underhill [61], these Pleistocene rocks were deposited in a shallow marine basin that received substantial clastic material from river sources, with the primary source being the thrust front of the Ainos Mountain.

### 4.2. Corfu Island

North of the Agios Georgios Pagon (Figure 7), along the coastal area of Arillas, there is an extensive sequence of Miocene deposits that covers a substantial area, approximately 1 km in length, extending to the cape of Agios Stefanos (Figure 7b) marking the Miocene-Pliocene boundary.

Stratigraphically, the lower deposits (South Arillas section) exhibit shifts between episodes of low and high energy, primarily featuring silt and sand, respectively (Figures 8a and 9a). This high to low energy deposits swift is characterized by rare basal conglomeratic beds evolving upwards to sandstone and mudstone beds (Figure 9b), showing a characteristic finning, and thinning upwards trend, whereas the sandy beds exhibit wavy laminations (Figure 9c). According to Tserolas et al. [62,63], below the above sequence, there is a gradual transition from flaser to wavy, and further to lenticular bedding, documenting an environment influenced by tides [62,63].

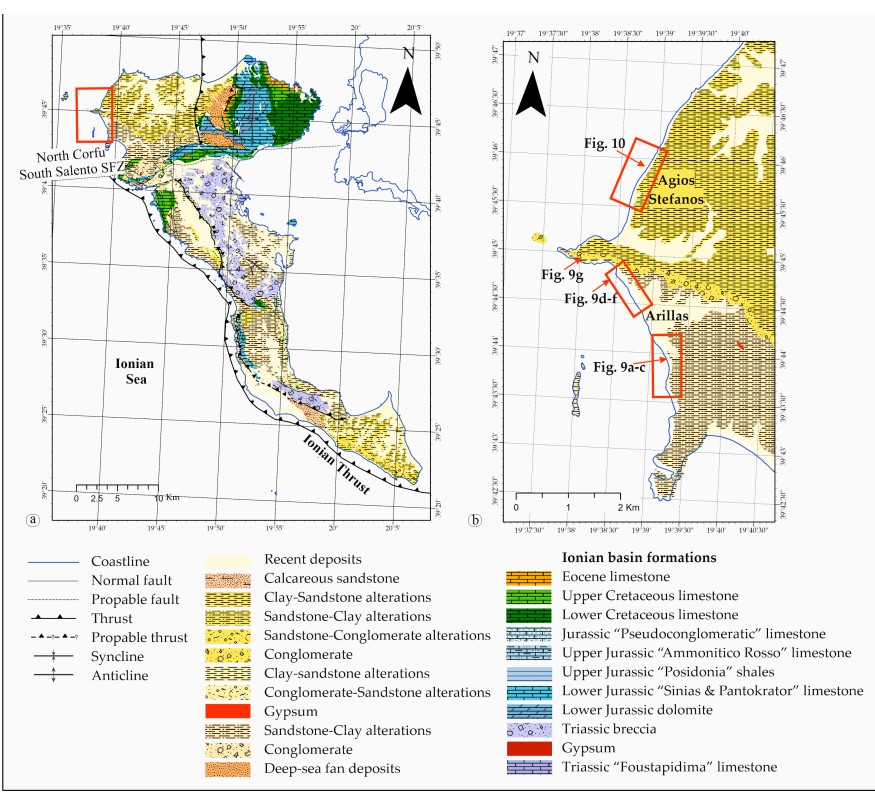

**Figure 7.** (**a**,**b**) Geological map of Corfu Island. The red frames mark the studied areas (modified after [62]).

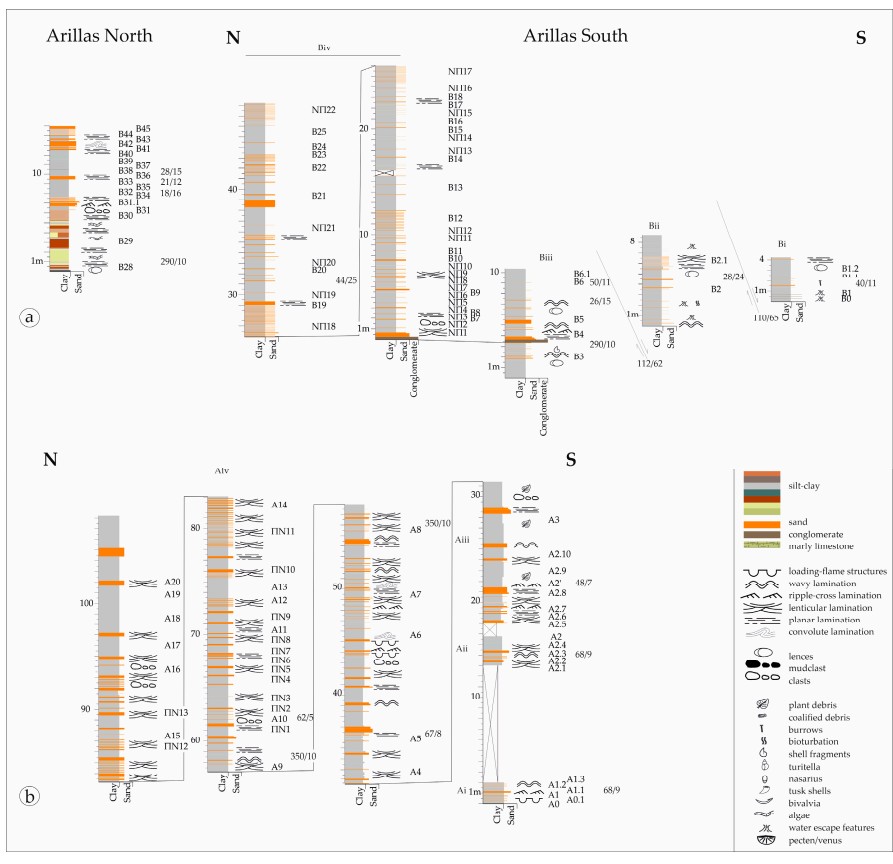

**Figure 8.** Representative stratigraphic logs (Supplementary Materials Figures S5–S7) of (**a**) Arillas and (**b**) Agios Stefanos studied sections.

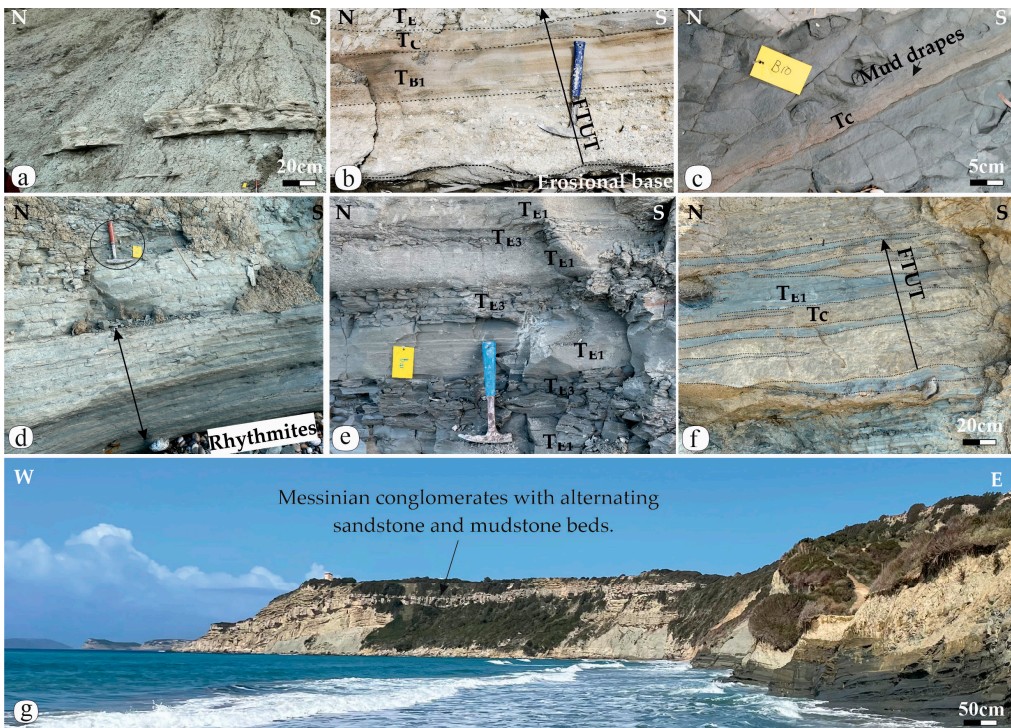

**Figure 9.** (**a**) Photo of South Arillas section depicting a conglomeratic bed in a mud-rich sequence; (**b**) fining and thinning upward trend (FTUT) of flow that evolve from conglomerate at bottom to parallel laminated sandstone ($T_{B1}$), wavy laminated sandstone ($T_C$), and mudstone ($T_{E1}$); (**c**) ripple-cross laminated sandstone ($T_C$); (**d**) sand and silt alterations; (**e**) alterations of parallel laminated ($T_{E2}$) and structureless mudstone ($T_{E1}$); (**f**) interbedded lenticular sand layers ($T_C$) within the mudstone ($T_{E1}$) sequence; (**g**) panoramic view of the Messinian conglomerates at the north Arillas bay.

In the northern part of the beach area (Figures 7b and 8a), there is a sequence characterized by alternating layers of sand and silt (Figures 8b and 9a,d,e), with an average thickness of approximately 15 m. Unlike deposits formed under slack conditions with a high concentration of suspended sediment, these deposits lack a distinct and thick mud layer. These rhythmites (Figures 8b and 9d–f) are characterized by flat, parallel-laminated bedding, with alternating deposition of muddy materials on neaps and sandy material on springs [64]. The contacts between the laminations are sharp but not erosional. Additionally, interbedded lenticular sand layers are observed within the sequence (Figure 9f). Stratigraphically higher the Miocene-Pliocene boundary is documented (Figure 9g) [62,63].

On the northern side of Agios Stefanos bay (Figures 7b, 8b and 10), there are distinct Pliocene phases (Figure 10a). The first phase, with a total thickness of around 20 m, features slightly thicker sandy layers and a higher sand/silt ratio that exhibit cyclic patterns (rythmites) (Figure 10a). This phase is characterized by distinct grading structures and parallel and wavy laminations (Figure 10b), with variations in grain size [62,64]. This alteration could be attributed to variations between calm water states (characterizing the wavy laminated sandstone) [65–67] and elevated energy levels could lead to swift sediment deposition (structureless or parallel laminated sandstone) [68–71]. This quick sedimentation is further justified by the presence of organic rich layers on the bottom of those beds (Figure 10c) because of the quick burial and preservation of the organic material. The followed phase, stratigraphically up-section, is notable for the prevalence of sandy layers in rhythmic alternation with thin layers of gray silt. The sandy layers exhibit hummocky (Figure 10d) and flaser laminations (Figure 10e). Towards the lower portions of the section, there are layers of very coarse sand to gravel, each approximately 10 to 20 cm in thickness or display convoluted lamination (Figure 10f) suggesting high-energy conditions, covered by mud drapes (Figure 10f). Clasts of silt are also present at the base of these

sandy layers. In the northwest end of the bay, carboniferous bedrock is exposed [62,65]. These deposits remain largely undisturbed, except for a few small normal faults with minor displacement, mostly oriented perpendicular to the general bedding. Based on the dip of the bedding (slightly eastward) and the elevation of each cliff, the estimated total thickness exceeds 400 m.

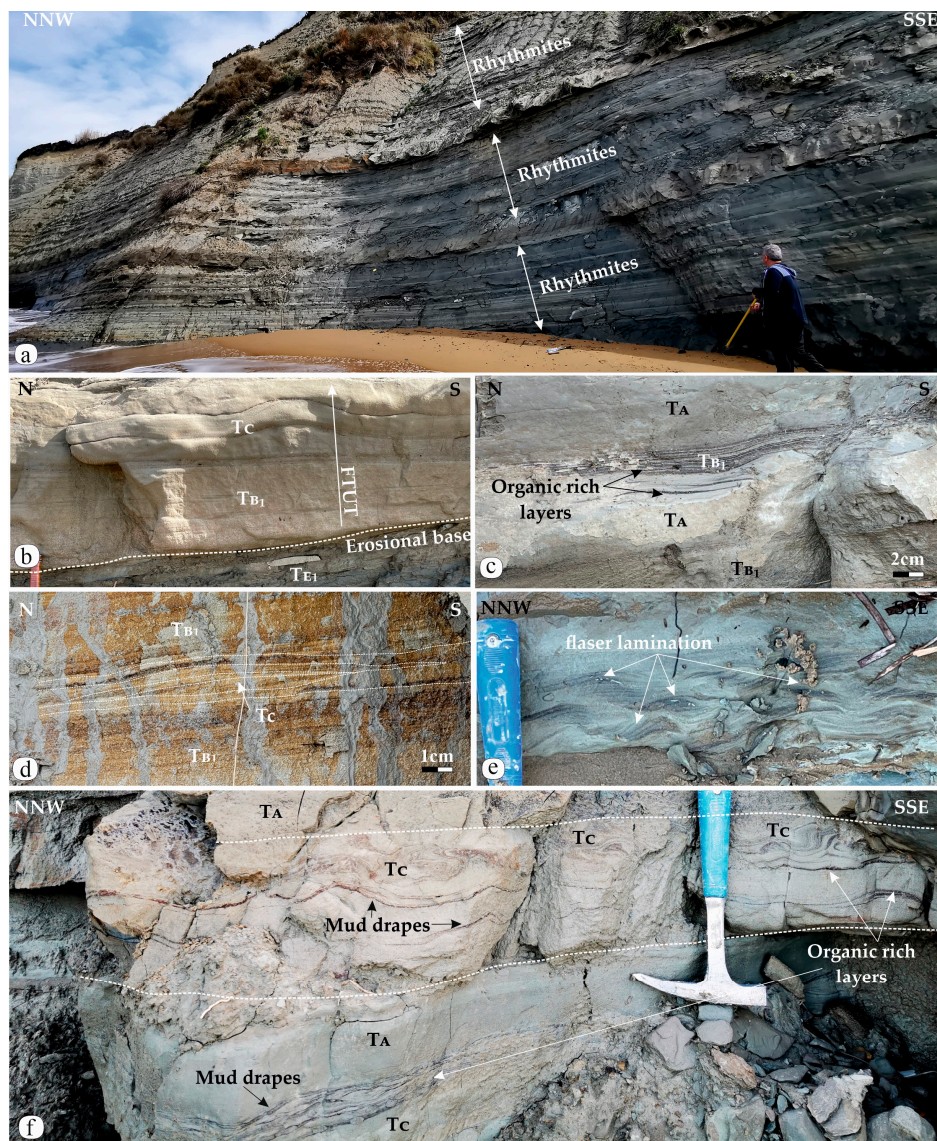

**Figure 10.** (**a**) Rhythmically sandstone–mudstone alterations of Agios Stefanos section; (**b**) parallel laminated sandstone (T$_{B1}$) evolving to wavy laminated sandstone (T$_C$), Abbreviations FTUT: Finning and Thinning upward trend; (**c**) organic rich layers between structureless sandstone beds (T$_A$); (**d**) hummocky cross-lamination (T$_C$); (**e**) flaser laminated sandstone; (**f**) structureless sandstone beds (T$_A$) and convoluted sandstone beds (T$_C$) with mud drapes producing flaser lamination.

## 5. Biostratigraphic Analysis

Planktonic foraminifera have been utilized for dating the succession due to their high abundance and their capability to facilitate interregional correlation. The biostratigraphic results were evaluated in relation to the stratigraphic position of the samples and the sections. A table of the estimated ages in each studied sequence is presented (Table 1).

**Table 1.** Age determination and paleoenvironmental implications of the studied sections.

| Studied Area | Section | Age Determination | Characteristic Pl. Foraminifera |
|---|---|---|---|
| Kefalonia Island | | | |
| Xi beach | X | Piacenzian | *Globoconella inflata, Globigerinoides bollii, Globigerinoides extremus* |
| Koutala beach | K | Upper Piacenzian to Gelasian | *Globigerinoides tenellus, Globigerinoides extremus* |
| Matzavinata | MA | Upper Zanclean to Gelasian | *Globigerinoides extremus, Globoturborotalita rubescens* |
| Chavdata | CH | Zanclean to Gelasian | *Globigerinoides elongatus* |
| Corfu Island | | | |
| Arillas South | B | Tortonian to Messinian | *Globorotalia plesiotumida, Globorotalia tumida, Globoturborotalita nepenthes, Globoturborotalita apertura, Globigerinoides conglobatus* |
| Arillas North | | at least Middle Miocene or younger age | *Cyprideis torosa* |
| Agios Stefanos | A | Zanclean | *Globigerinoides conglobatus, Globorotalia margaritae, Neogloboquadrina dutertei* |

*5.1. Kefalonia Island*

In Kefalonia, the stratigraphic arrangement of Xi Beach unveils a westward-sloping sequence, with older sediments situated to the east and progressively younger deposits to the west. For instance, Xi Beach (samples X10.1, X21, X29, X33, X39, X42, X48, and X49) suggests a Piacenzian age, with the coexistence of characteristic foraminiferal fossils like *Globoconella inflata*, *Globigerinoides bollii* and *Globigerinoides extremus*. These are deep-sea settings with depths ranging from 50 to 200 m, as indicated by the abundant presence of characteristic benthic foraminifera such as *Cibicidoides mundulus*, *Sphaeroidina bulloides*, *Bolivina* spp., and *Bulimina* spp. In these environments, there is a prevalence of low oxygen levels, signifying high organic matter content, as evidenced by the abundance of *Uvigerinidae* and *Cassidulina carinata*. Noteworthy are occurrences of upward currents (upwelling), exemplified in samples X29 and X32 (mid-section of the sequence), where planktonic foraminifera like *Globigerina bulloides* and *Orbulina* dominate.

Moreover, in Koutala Beach, the existence of planktonic foraminifers such as *Globigerinoides tenellus* and *Globigerinoides extremus* indicates Upper Piacenzian to Gelasian age. This part of the sequence (sample K0) was deposited in an upper downslope environment, with dominant planktonic foraminifers *Trilobatus trilobus* and *Globigerinoides* spp. indicating warm waters with salinity variations. The coexistence of benthic foraminifera adapted to low-oxygen conditions (e.g., *C. carinata*), with those thriving in well-oxygenated environments (e.g., *Globocassidulina subglobosa*, *Cibicides refulgens*, etc.), suggests fluctuations in oxygen levels and underscores seasonal variations. The upper section of the sequence (sample K8) showcases sediment deposition in deep-sea environments, specifically in upper and middle bathyal zones, as indicated by the presence of benthic foraminifera such as *Sphaeroidina bulloides* and *C. mundulus*. Moreover, the presence of *Uvigerinidae*, typically found in shallower environments like the outer continental shelf and upper downslope, implies the prevalence of dysoxic bottom conditions. It is noteworthy that planktonic foraminifera are conspicuously absent.

The existence of planktonic foraminifera species *Globigerinoides extremus* and *Globoturborotalita rubescens* (in samples MA1 and MA14) signifies the Upper Pliocene–Lower Pleistocene age (Upper Zanclean to Gelasian) for the Matzavinata sequence. The sedimentation of this sequence occurred in the profound depths of the deep sea (as determined by the abundant presence of *C. mundulus*), characterized by low oxygen conditions (*C. carinata*, *Uvigerinidae*).

The existence of the planktonic foraminifera *G. elongatus* (in samples CH1 and CH6) suggests an Upper Pliocene–Lower Pleistocene age (Upper Zanclean-Gelasian) age for Chavdata section. The sedimentation of this sequence occurred in deep marine environ-

ments. The composition of the burial community hints at deep conditions (200–300 m or possibly more, indicated by the presence of *Planulina ariminensis*, *Rosalina orbicularis*, etc.). The increased presence of benthic foraminifera indicative of low oxygen suggests the potential existence of such environments at shallower depths. The coexistence of these foraminifera with those adapted to well-oxygenated conditions (e.g., *Cibicides refulgens*, etc.) likely reflects seasonal variations.

### 5.2. Corfu Island

The age of the Arillas lower stratigraphic section (in the southern part of the gulf), primarily due to the presence of the planktonic foraminifers *Globorotalia plesiotumida*, *Globorotalia tumida*, *Globoturborotalita nepenthes*, *Globoturborotalita apertura*, *Globigerinoides conglobatus* is Tortonian to Messinian. The absence of benthic foraminifera in sample B1.2 probably indicates anoxic bottom conditions. The depositional environments likely experienced frequent fluctuations in depth and surface water temperature, as evidenced by some samples, such as B6.1, showing a predominance of colder-temperate species like *Globigerina bulloides*, combined with an increase in *Globorotaliids* (deep dwellers). The northern part of Arillas sequence includes many barren samples. This, combined with the presence of samples containing significant quantities of the typical brackish water ostracod *Cyprideis torosa* and the foraminifera *Ammonia tepida* (sample B32), indicates the terrestrial origin of the sequence. This combination is characteristic of restricted lagoon systems with limited communication with the sea. The age of the sequence cannot be determined as no characteristic fossils were found. It is noted, however, that *C. torosa* evolved after the Mid-Miocene.

The Agios Stefanos stratigraphic section is characterized as Zanclean, based on the presence of *Globigerinoides conglobatus*, *Globorotalia margaritae* and *Neogloboquadrina dutertei* and the stratigraphic position of the samples. The sediments in this section have been deposited in a deep marine environment (depth 100–300 m) with transported shells from the continental slope. The deep-dwelling species *Cibicidoides mundulus* dominates, and there are indications of low oxygen levels.

### 5.3. Age Determination for Kefalonia and Corfu Islands

The integrated analysis of planktonic and benthic foraminifera species allowed for robust age determinations of sedimentary samples from both Kefalonia and Corfu Island. The presence of different *Globigerinoides* spp. in Kefalonia samples suggests a Zanclean to Gelasian age. For Corfu Island samples, the estimated ages vary from Tortonian/Messinian to Zanclean. Samples from Arillas area bear a Tortonian to Messinian age while samples from the Agios Stefanos section indicated a Zanclean age. It is noteworthy that all the samples contained significant numbers of species indicative of older ages such as *Globigerinoides kenneti*, *Globoconella puncticulata*, *Globigerinoides subquadratus* (Kefalonia samples) and *Globigerinatella*, *Globigerina pseudopraebulloides* and *Globorotalia archaeomenardii* (indicative of Burdigalian ages for Kerkyra samples). These species are considered transported.

## 6. Geochemical and Sedimentological Analysis

### 6.1. Calcium Carbonate (CaCO₃)

Laboratory analysis of calcium carbonate ($CaCO_3$) shows that the percentage ranges from 12.22% to 39.27% as seen in Table 2 and graphs (Figure 11 and Supplementary Materials Table S1). Despite the deviation of the maximum from the minimum value, uniformity is observed in the distribution of $CaCO_3$, which tends to be concentrated around 26.71% (TM). The average percentage of the Corfu (TC) samples is 30.90% and the average percentage of the Kefalonia (TK) samples is 24.60%.

**Table 2.** Average % calcium carbonate content of the studied sections.

| Studied Area | | Section | Calcium Carbonate ($CaCO_3$) | | |
|---|---|---|---|---|---|
| | | | Min | Max | Aver. |
| Kefalonia Island | Xi beach | X | 16.00 | 34.72 | 24.27 |
| | Xi beach sand | | | 17.87 (1 sample) | |
| | Koutala beach | K | 20.28 | 33.66 | 25.36 |
| | Matzavinata | MA | 19.1 | 34.6 | 25.7 |
| | Chavdata | CH | 12.22 | 35.81 | 24.14 |
| Corfu Island | Arillas | B | 17.76 | 39.27 | 31.86 |
| | Agios Stefanos | A | 22.86 | 37.45 | 29.96 |


(**A**) (**B**)

**Figure 11.** $CaCO_3$ percentage of (**A**) Kefalonia Island samples; (**B**) Corfu Island samples. Abbreviations MC: the mean value of Corfu Island samples; MT: the mean value of the overall samples and MK: the mean value of Kefalonia Island samples.

### 6.1.1. Kefalonia Island

The concentration of $CaCO_3$ in Kefalonia Island (Xi beach, Koutala beach, Chavdata, and Matzavinata) was analyzed to investigate cycles of sedimentation. The $CaCO_3$ percentage in Xi beach ranged from 16.00% to 34.72% (average 24.27%). Samples showed variation in $CaCO_3$ concentrations compared to MT. In the first sedimentation cycle (samples X71-X42), the maximum value is X47 (33.1%), and the minimum value is in sample X71 (17.87%). In the second sedimentation cycle (samples X41-X32), the maximum value of $CaCO_3$ is X37 (27.1%), while the minimum value is in sample X41 (19.07%). In the third sedimentation cycle (samples X31-X29), the maximum value is X30 (34.72%), and the minimum value is in sample X31 (16.00%). In the fourth sedimentation cycle (samples X28-X23), the maximum value of $CaCO_3$ is in X24 (34.5%), and the minimum value is in sample X28 (17.05%). In the fifth sedimentation cycle (samples X22-X21), the maximum value occurs in X21 (34.11%), while the minimum value is in X22 (20.25%). In the sixth sedimentation cycle (samples X19-X17), the maximum value of $CaCO_3$ is found in X17 (24.8%), and the minimum value is in sample X19 (19.75%). In the last sedimentation cycle (samples X16-X10), the maximum value of $CaCO_3$ appears in sample X11 (34.41%), while the minimum value is observed in sample X16 (18.35%). The average $CaCO_3$ content shows a general increasing trend from the first to the seventh sedimentation cycle.

The $CaCO_3$ percentage in Koutala beach ranged from 20.28% to 33.66% (average 25.36%) with periodic increase and decrease, indicating cycles. Samples showed variation in $CaCO_3$ concentrations compared to MT. In the first sedimentary cycle (samples K0-K3), the maximum $CaCO_3$ value was observed in sample K1 (33.66%), while the minimum value was observed in sample K3 (20.72%). In the second sedimentary cycle (samples K4-K8), the maximum value was observed in sample K6 (27.16%), while the minimum value was observed in sample K8 (20.28%). The average $CaCO_3$ content showed a general upward decreasing trend from the first to the second sedimentary cycle.

In Matzavinata, the $CaCO_3$ percentage ranged from 19.1% to 34.6% (average 25.7%). Samples showed variation in $CaCO_3$ concentrations compared to MT. In the first sedi-

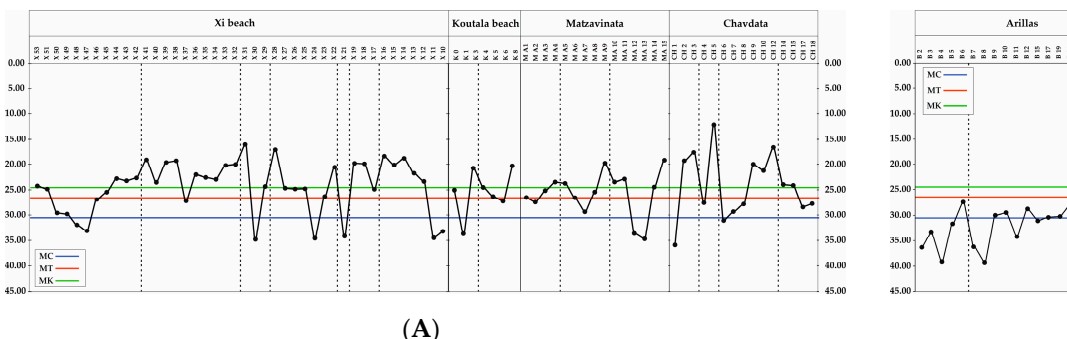

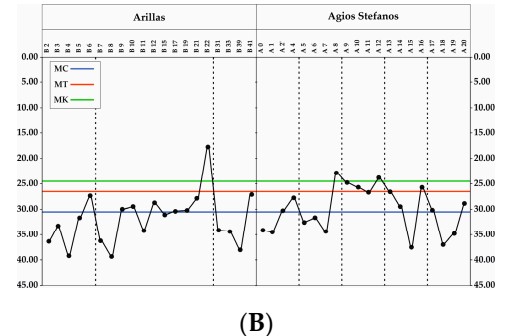

mentary cycle (samples MA1-MA4), the maximum $CaCO_3$ value was observed in sample MA2 (27.34%), while the minimum value was observed in sample MA4 (23.43%). In the second sedimentary cycle (samples MA5-MA9), the maximum value was observed in sample MA7 (29.29%), while the minimum value was observed in sample MA9 (19.72%). The average $CaCO_3$ content showed a general decreasing trend from the first to the third sedimentary cycle.

Finally, in Chavdata, the $CaCO_3$ percentage varied from 12.22% to 35.81% (average 24.14%). Samples showed variation in $CaCO_3$ concentrations compared to MT. In the first sedimentary cycle (samples CH1-CH3), the maximum $CaCO_3$ value was observed in sample CH1 (35.81%), while the minimum value was observed in sample CH3 (17.56%). In the second sedimentary cycle (samples CH4-CH5), the maximum value was observed in sample CH4 (27.45%), while the minimum value was observed in sample CH5 (12.22%). The average $CaCO_3$ content showed a general decreasing trend from the first to the fourth sedimentary cycle.

### 6.1.2. Corfu Island

The $CaCO_3$ percentage in Arillas ranges from 17.76% to 39.27% (average 31.86%). Most of the samples show enrichment in $CaCO_3$ compared to MT, with only one sample in the upper part of the stratigraphic column showing deficiency. The periodic fluctuation of $CaCO_3$ concentration around a certain value indicates sedimentation cycles. In the first cycle (samples B2-B6), the maximum value of $CaCO_3$ is observed in sample B4 (39.12%), while the minimum value is observed in sample B6 (27.45%). In the second cycle (samples B7-B22), the maximum value of $CaCO_3$ is observed in sample B8 (39.27%), while the minimum value is observed in sample B22 (17.76%). In the third cycle (samples B31-B41), the maximum value is observed in sample B39 (37.95%), while the minimum value is observed in sample B41 (27.12%). The average $CaCO_3$ content shows an overall increasing trend in all sedimentary cycles.

Similarly, in the Agios Stefanos, the $CaCO_3$ percentage ranges from 22.86% to 37.45% (average 29.96%), with most samples showing an enrichment in $CaCO_3$ compared to MT, except for some samples in the middle part of the stratigraphic column. In the first cycle (samples A0-A4), the maximum value of $CaCO_3$ was observed in sample A1 (34.50%), while the minimum value is observed in sample A4 (27.77%). In the second cycle (samples A5-A8), the maximum value of $CaCO_3$ was observed in sample A7 (34.39%), while the minimum value was observed in sample A8 (22.86%). In the third cycle (samples A9-A12), the maximum value was observed in sample A11 (26.68%), while the minimum value was observed in sample A12 (23.90%). In the fourth cycle (samples A13-A16), the maximum value of $CaCO_3$ was observed in sample A15 (37.45%), while the minimum value was observed in sample A16 (25.72%). In the fifth cycle (samples A17-A20), the maximum value of $CaCO_3$ was observed in sample A18 (36.86%), while the minimum value was observed in sample A20 (28.83%). The average $CaCO_3$ content shows an overall decreasing trend from the first to the third sedimentary cycle, which then switches to an overall increasing trend from the third to the fifth sedimentary cycle.

### 6.2. Total Organic Carbon Content (TOC)

Laboratory analysis of TOC shows that the percentage ranges from 0.10% to 0.77%, as seen in Table 3 and graphs (Figure 12 and Supplementary Materials Table S1). Despite the deviation of the maximum from the minimum value, uniformity is observed in the distribution of organic carbon, which tends to be concentrated around 0.32% (TM). The average percentage of Corfu (TC) samples is 0.30% and the average percentage of Kefalonia (TK) samples is 0.33%.

**Table 3.** Average % total organic carbon content of the studied sections.

| Studied Area | Section | Total Organic Carbon (Corg) | | |
| --- | --- | --- | --- | --- |
| | | Min | Max | Aver. |
| Kefalonia Island | X | 0.13 | 0.77 | 0.42 |
| Xi beach | | | | |
| Xi beach sand | | | 0.13 (1 sample) | |
| Koutala beach | K | 0.19 | 0.62 | 0.36 |
| Matzavinata | MA | 0.10 | 0.29 | 0.18 |
| Chavdata | CH | 0.13 | 0.47 | 0.23 |
| Corfu Island | B | 0.11 | 0.45 | 0.27 |
| Arillas | | | | |
| Agios Stefanos | A | 0.18 | 0.58 | 0.33 |

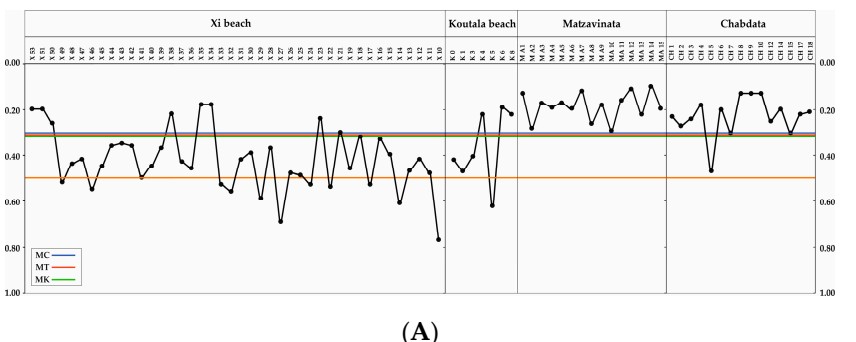
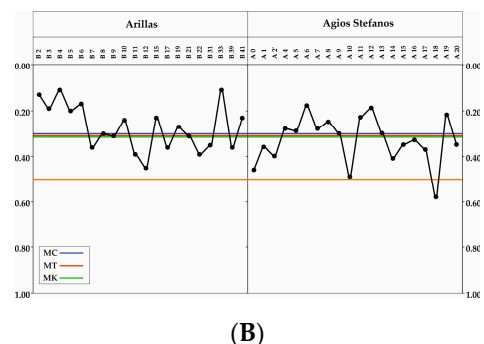

(**A**)   (**B**)

**Figure 12.** TOC percentage of (**A**) Kefalonia Island samples and (**B**) Corfu Island samples. Abbreviations MC: the mean value of Corfu Island samples; MT: the mean value of the overall samples and MK: the mean value of Kefalonia Island samples.

6.2.1. Kefalonia Island

The TOC percentage ranges in Xi Beach from 0.13% to 0.77% (average 0.42%). The samples show a variation, with the samples being mostly enriched in TOC compared to MT, while from the base to the top of the stratigraphic column there is a general decreasing trend. The maximum value of TOC occurs in sample X10 (0.77%), while the minimum value occurs in sample X71 (0.13%). In addition, the concentration of twelve samples (X10, X14, X17, X22, X24, X27, X29, X32, X33, X41, X46 and X49) exceeds 0.5%.

The TOC percentage varies in Koutala Beach from 0.19% to 0.62% (average 0.36%). Samples show a variation, with samples being enriched or depleted in TOC compared to MT. In addition, its concentration in one sample (K5) exceeds 0.5%. The maximum value of TOC occurs in sample K5 (0.62%), while the minimum value occurs in sample K6 (0.19%).

The TOC percentage varies in Matzavinata from 0.10% to 0.29% (average 0.18%). The samples appeared to have a lower concentration of TOC compared to MT, while no sample exceeds 0.5%. The maximum value of TOC occurred in sample MA10 (0.29%), while the minimum value occurred in samples MA12 and MA14 (0.11% and 0.10%, respectively).

The TOC percentage ranged in Chavdata from 0.13% to 0.47% (average 0.23%). The samples appeared to have a lower concentration of TOC compared to MT (only sample CH5 appears to have a higher concentration), while no sample exceeded 0.5%. The maximum value of TOC occurred in sample CH5 (0.47%), while the minimum value occurred in samples CH8, CH9 and CH10 (0.13%).

6.2.2. Corfu Island

The TOC percentage ranged in Arillas from 0.11% to 0.45% (average 0.27). Most samples were poor in TOC compared to MT, while only seven samples were enriched. Furthermore, no sample exceeded 0.5%. The maximum value of TOC occurred in sample B12 (0.45%), while the minimum value occurred in samples B4 and B33 (0.11%).

The TOC percentage ranges in Agios Stefanos from 0.18% to 0.58% (average 0.33%). The samples appear to have a periodic fluctuation in their concentration of TOC compared

to MT. Furthermore, its concentration in one sample (A18) exceeded 0.5%, while sample A10 (0.49%) barely reached this value. The maximum value of TOC occurs in sample A18 (0.58%), while the minimum value occurs in samples A6 and A12 (0.18% and 0.19%, respectively).

### 6.3. Comprehensive Analysis of Elements, Mineralogical Composition, and Physical Properties

The extensive analysis conducted on the studied mudstones in Kefalonia and Corfu Islands is throughout presented in [17], providing valuable insights into the presence of major and trace elements, mineralogical composition, and physical characteristics.

### 6.3.1. XRF Analysis

The concentrations of the studied elements generally range within the permissible limits, with 9 to 14 toxic elements presenting values below the upper maximum limits for both islands mudstone. Nickel (Ni) exceeds the permissible limits in almost all samples (ranging from 70 to 190 mg/kg and 170 to 240 mg/kg for Kefalonia and Corfu Island, respectively), potentially causing allergic contact dermatitis. Toxic elements that exceed the upper allowed limits are also strontium (Sr) ranging from 160 to 540 mg/kg and 270 to 555 mg/kg for Kefalonia and Corfu Island, respectively; gallium (Ga) ranging from 9 to 46 mg/kg and 19 to 44 mg/kg for Kefalonia and Corfu Island, respectively; molybdenum (Mo) ranging from 1 to 17 mg/kg and 1 to 11 mg/kg for Kefalonia and Corfu Island, respectively and tellurium (Te) ranging from <1 to 2 mg/kg and 1.5 to 2 mg/kg for Kefalonia and Corfu Island, respectively.

### 6.3.2. XRPD Analysis

The X-ray powder diffraction (XRPD) analysis revealed variations in the concentrations of main elements, showcasing correlations and mineralogical compositions. The results document concentrations for $SiO_2$ ranging from 32.8 to 47.97% and 32.8 to 45.2%, $Al_2O_3$ from 8.88 to 14.27% and 9.29 to 12.26%, $Fe_2O_3$ from 4.33 to 8.12% and 5.89 to 7.86%, MgO from 0.09 to 0.18% and 0.11 to 0.18%, CaO from 13.56 to 30.25% and 16.8 to 30.3%, $Na_2O$ from 0.43 to 1.73% and 0.43 to 1.25%, $K_2O$ from 1.67 to 2.97% and 2.02 to 2.62%, $TiO_2$ from 0.55 to 0.79% and 0.55 to 0.69%, $P_2O_5$ from 0.08 to 0.14% and MnO from 0.09 to 0.18% and 0.11 to 0.18%, for Kefalonia and Corfu Islands, respectively. $SiO_2$ exhibits positive correlations with $Al_2O_3$, $TiO_2$, $K_2O$, $Na_2O$, and $P_2O_5$, while exhibiting a negative correlation with CaO. The presence of clay minerals, smectite, quartz, calcite, dolomite, and plagioclase varied across the samples. Mineralogical analysis documents that quartz ranges from 1 to 12% and 1 to 19% and plagioclase ranges from 1 to 12% and 1 to 19% for Kefalonia and Corfu Islands, respectively. Furthermore, the calcite percentage ranges from 1 to 12% and 1 to 19%, and the dolomite percentage ranges from 1 to 12% and 1 to 19% for Kefalonia and Corfu Islands, respectively. Illite, Chlorite and Kaolinite range from 2 to 12%, 2 to 8% and 2 to 7% for Kefalonia Island and 3 to 12%, 7 to 14% and 2 to 8% for Corfu Island, respectively, indicating that clay minerals range from 5 to 22% and 13 to 31% for Kefalonia and Corfu Islands, respectively.

### 6.3.3. pH Determination

Samples collected from Kefalonia island exhibit pH levels ranging from 7.8 to 9.6, and their mean value is 8.75. On the other hand, samples from Corfu Island show pH levels ranging from 8.1 to 9.6, with a mean value of 8.6. Consequently, the mudstones can be described as ranging from slightly alkaline to very strongly alkaline, with most samples demonstrating a strongly alkaline nature.

### 6.3.4. Plasticity Determination

Plasticity analysis of the samples collected from Kefalonia island exhibit liquid limit (LL) ranging from 27 to 78%, with an average of 42.45%, and plasticity index (PI) ranging from 1 to 62% with an average of 18.73%. On the other hand, samples from Corfu Island

show LL ranging from 32 to 54% with an average of 44.2%, and PI ranging from 10 to 29% with an average of 16.48%. Conceivably, most of the samples based on the Casagrande plasticity diagram are characterized as low plasticity clays, while only two are projected in the field of high-plasticity muds.

### 6.3.5. Specific Surface Area Determination

Specific surface area analysis of the samples collected from Kefalonia island exhibit values ranging from 6 to 29 m$^2$/gr with an average of 16.55 m$^2$/gr. The values varied across locations, with Xi Beach and Kefala Beach showing similarities, Matzavinata having slightly higher values, and Chavdata exhibiting the highest averages. On the other hand, samples from Corfu Island exhibit values ranging from 9 to 28 m$^2$/gr with an average of 19.84 m$^2$/gr. In Corfu Island, Arillas Beach samples generally have larger specific surface areas compared to Agios Stefanos.

### 6.3.6. Thermal Analysis

Thermal analysis indicated significant weight loss initiates around 670 °C, because of the calcination and decomposition of calcium carbonate (CaCO$_3$) in samples from Matzavinata (Kefalonia) and Agios Stefanos (Corfu) areas.

### 6.4. Grain Size Analysis

The sediment samples demonstrate a diverse mixture of various grain sizes (Table 4 and Supplementary Materials Table S2). The average composition of the Kefalonia samples consists of 3.46% sand, 57.80% silt, and 38.74% clay. The same tendency is documented on the average composition of the Corfu samples consisting of 9.32% sand, 54.75% silt, and 35.94% clay.

**Table 4.** Average % grain size distribution of the studied sections.

| Studied Area | Grain Size | Clay | | | Silt | | | Sand | | |
|---|---|---|---|---|---|---|---|---|---|---|
| | | Min | Max | Aver. | Min | Max | Aver. | Min | Max | Aver. |
| Kefalonia Island | Xi beach | 7.34 | 51.28 | 37.09 | 46.96 | 88.63 | 59.04 | 0.80 | 19.07 | 3.87 |
| | Koutala beach | 4.88 | 49.59 | 31.49 | 49.81 | 87.85 | 65.45 | 0.60 | 7.28 | 3.06 |
| | Matzavinata | 3.78 | 48.86 | 37.3 | 50.03 | 90.72 | 58.60 | 1.11 | 11.16 | 4.10 |
| | Chavdata | 33.89 | 53.84 | 48.48 | 43.53 | 64.89 | 49.98 | 0.57 | 3.94 | 1.85 |
| Corfu Island | Arillas | 8.73 | 55.90 | 28.79 | 27.04 | 84.61 | 58.54 | 3.46 | 44.87 | 12.67 |
| | Agios Stefanos | 0.18 | 56.24 | 43.44 | 39.78 | 98.89 | 50.77 | 0.93 | 10.97 | 5.79 |

#### 6.4.1. Kefalonia Island

For the Xi beach section, the sand content ranges from 0.80% to 19.07% (average 3.87%), silt content ranges from 46.96% to 88.63% (average 59.04%), and clay content ranges from 7.34% to 51.28% (average 37.09%). For the samples in the Koutala beach section, sand ranges from 0.60% to 7.28% (average 3.06%), the silt content ranges from 49.81% to 87.85% (average 65.45%), and the clay content ranges from 4.88% to 49.59% (average 31.49%). For the samples in the Matzavinata section, the sand content ranges from 1.11% to 11.16% (average 4.10%), silt content ranges from 50.03% to 90.72% (average 58.60%), and clay content ranges from 3.78% to 48.86% (average 37.30%). In the Chavdata section, sand content ranges from 0.57% to 3.94% (average 1.85%), silt content ranges from 43.53% to 64.89% (average 49.98%), and clay content ranges from 33.89% to 53.84% (average 48.48%). Across all these samples, silt consistently emerged as the primary component, followed by clay, with sand content being the lowest component in terms of composition.

#### 6.4.2. Corfu Island

For the Arillas section, the sand content ranges from 3.46% to 44.87% (average 12.67%), silt content ranges from 27.04% to 84.61% (average 58.54%), and clay content ranges from

8.73% to 55.90% (average 28.79%). In the case of Agios Stefanos, the sand content ranges from 0.93% to 10.97% (average 5.79%), silt content ranges from 39.78% to 98.89% (average 50.77%), and clay content ranges from 0.18% to 56.24% (average 43.44%). Across all these samples, silt consistently emerges as the primary component, followed by clay, and sand content being the lowest component in terms of composition.

### 6.5. Cumulative Frequency Curves

Phi scales are plotted on a probability scale as cumulative frequency curves (Supplementary Materials Figures S8 and S9), providing insights into different modes of sediment transportation, deposition, and their significance in sedimentation [72]. Sorting of the samples can be determined by examining the slope of the middle portion of the curve [73].

In this context, poor sorting is represented by a wide and gently sloping curve, indicating low kinetic energy and velocity. This suggests a significant variation in particle sizes, leading to a diverse range of grain sizes. Conversely, a steep slope on the curve is indicative of good sorting, reflecting high kinetic energy, high velocity, and uniformity in particle size with minimal variation. The studied samples are primarily fine- to very fine-grained. However, considering the moderate gradient of the cumulative frequency curves, these samples generally exhibit low uniformity in particle size, implying inadequate sorting and a sedimentary environment characterized by low kinetic energy and velocity.

### 6.6. Statistical Parameters

The results of a grain size analysis that has been processed with Origin Software are shown in Table 5 and Supplementary Materials Table S2. Statistical parameters, including mean, sorting, kurtosis, and skewness, were derived from the outcomes of grain size analysis. These parameters were then utilized to generate graphs using Grapher software v. 16.2.354.

**Table 5.** Statistical parameters of the studied sections.

| Studied Area | | Section | Mean Size ($M_z$) | | | Sorting ($\sigma_1$) | | | Skewness ($SK_1$) | | | Kurtosis ($K_G$) | | |
|---|---|---|---|---|---|---|---|---|---|---|---|---|---|---|
| | | | Min | Max | Aver. | Min | Max | Aver. | Min | Max | Aver. | Min | Max | Aver. |
| Kefalonia Island | Xi beach | X | 5.54 | 8.82 | 8.04 | 1.09 | 3.50 | 2.76 | 0.11 | 0.66 | 0.43 | 0.65 | 2.63 | 1.03 |
| | Xi beach sand | | 2.28 | 2.86 | 2.51 | 0.31 | 0.62 | 0.43 | −0.37 | −0.07 | −0.23 | 0.93 | 1.51 | 1.15 |
| | Koutala beach | K | 6.23 | 8.80 | 7.41 | 1.19 | 2.88 | 2.20 | 0.37 | 0.73 | 0.45 | 0.61 | 1.28 | 1.01 |
| | Matzavinata | MA | 6.81 | 8.72 | 8.08 | 1.07 | 3.19 | 2.78 | 0.31 | 0.56 | 0.42 | 0.66 | 1.49 | 1.03 |
| | Chavdata | CH | 7.25 | 8.99 | 8.60 | 2.05 | 2.92 | 2.73 | 0.27 | 0.42 | 0.35 | 0.63 | 0.82 | 1.62 |
| Corfu Island | Arillas | B | 6.13 | 9.14 | 7.19 | 1.21 | 3.26 | 2.36 | −0.30 | 0.39 | 0.22 | 0.74 | 1.73 | 1.17 |
| | Agios Stefanos | A | 6.47 | 9.08 | 8.26 | 0.64 | 3.28 | 2.84 | −0.09 | 0.40 | 0.23 | 0.66 | 1.65 | 0.88 |

#### 6.6.1. Graphic Mean Size ($M_z$)

The mean size (Mz) is indicative of the average particle size within a sample. The calculated mean values for the studied sediments show similarities among the different studied areas. For Kefalonia Island sediments, the Mz range from 6.23∅ to 8.80∅ with an average value of 7.41∅ for Koutala Beach section and for Xi Beach section range from 5.54∅ to 8.82∅ with an average value of 8.04∅. Furthermore, for Chavdata section range from 7.25∅ to 8.99∅ with an average of 8.60∅ for and from 6.81∅ to 8.72∅ with an average of 8.08∅ for Matzavinata section. For Corfu Island sediments, the Mz range from 6.13∅ to 9.14∅ with an average value of 7.19∅ for Arillas section and from 6.47∅ to 9.08∅ with an overall mean value of 8.26∅ for Agios Stefanos section.

The samples of Kefalonia sediments indicate the predominance of clay, with a mean value of 8.10∅. On the other hand, the Mz values for the Corfu sediments suggest the prevalence of very fine silt, with a mean value of 7.71∅. Additionally, local variations may occur within these formations, ranging from moderate silt to clay. This general variation in Mz indicates fluctuations in energetic conditions during sediment deposition. Furthermore,

the dominance of fine-grained sediments and the absence of coarse-grained sediments suggest a moderately low-energy depositional environment [74].

### 6.6.2. Standard Deviation ($\sigma_1$)

The standard deviation ($\sigma_1$) values provide insights into the sorting or uniformity of grains, which, in turn, offer indications about the prevailing energy conditions during transportation and deposition. For Kefalonia Island sediments, the $\sigma_1$ values of Koutala Beach section range from 1.19$\varnothing$ to 2.88$\varnothing$, with an average of 2.20$\varnothing$ and Xi Beach section range from 1.09$\varnothing$ to 3.50$\varnothing$, with an average of 2.76$\varnothing$. Moreover, the $\sigma_1$ values range from 2.05$\varnothing$ to 2.92$\varnothing$, with a mean of 2.73$\varnothing$ for Chavdata section and from 1.07$\varnothing$ to 3.19$\varnothing$, with a mean of 2.78$\varnothing$ for Matzavinata section. For Corfu Island sediments, the $\sigma_1$ values range from 1.21$\varnothing$ to 3.26$\varnothing$, with a mean of 2.36$\varnothing$ for Arillas section and from 0.64$\varnothing$ to 3.28$\varnothing$, with a mean of 2.84$\varnothing$ for Agios Georgios section.

The size of the screening is influenced by the energy conditions and the duration of time during which currents or waves acted on the grain particles [75–77]. All the samples document poorly to very poorly sorted $\sigma_1$ which is also an indication of non-uniform energy condition prevailing at the time of deposition. In general, the poor sorting of sediments could be associated with deposition in environments influenced by storms or mudflows, as well as streams [74–77].

### 6.6.3. Graphic Skewness (SK$_1$)

The degree of skewness (SK$_1$) in cumulative curves provides information about the predominance of fine or coarse-grained fractions within sediment samples. For Kefalonia Island, the SK$_1$ values range from 0.37 to 0.73$\varnothing$, with an average of 0.45$\varnothing$, except for one sample (K3, 0.09$\varnothing$) for the Koutala beach section and for the Xi beach section, the SK$_1$ range from 0.11 to 0.66$\varnothing$, with an average of 0.43$\varnothing$. Furthermore, for the Chavdata section the SK$_1$ ranges from 0.27 to 0.42$\varnothing$, with an average of 0.35$\varnothing$ and for the Matzavinata section, except for one sample (MA15, −0.52$\varnothing$), the SK$_1$ range from 0.31 to 0.56$\varnothing$, with an average of 0.42$\varnothing$. For Corfu Island sediments, the SK$_1$ values for the Arillas section range from 0.08 to 0.39$\varnothing$, with an average of 0.22$\varnothing$, except two samples (B2, −0.10$\varnothing$, B22, −0.30$\varnothing$) and for the Agios Stefanos section, range from −0.09 to 0.40$\varnothing$, with an average of 0.23$\varnothing$.

Very fine skewed and fine skewed grains followed by near symmetrical skewed grains and coarse skewed grains dominate the samples equally. It is noteworthy that the absence of nearly symmetrical nature in the sediments, except for a few samples, may indicate extreme conditions such as tidal changes, wave breaking, and seasonal supply of detrital materials, as documented by Selvaraj and Ram Mohan [78].

### 6.6.4. Graphic Kurtosis (K$_G$)

Kurtosis (K$_G$) measures the degree of "peakedness" in a curve. K$_G$ is mainly characterized by very platykurtic behavior (thinner than normal tail), followed by platykurtic, very leptokurtic (thicker than normal tail), and mesokurtic behavior (equal thickness throughout the curve). For Kefalonia Island sediments the K$_G$ values for the Koutala beach section range from 0.61 to 1.28$\varnothing$, with an average of 1.03$\varnothing$, similar to Xi Beach samples (X) (values range from 0.65 to 2.63$\varnothing$, with an average value of 1.03$\varnothing$). For the Chavdata section, the K$_G$ values range from 0.63 to 1.62$\varnothing$, with an average value of 0.82$\varnothing$ and for Matzavinata section K$_G$ values range from 0.66 to 1.49$\varnothing$, with an average of 1.01$\varnothing$. For Corfu Island, Arillas section range from 0.74 to 1.73$\varnothing$, with an average value of 1.17$\varnothing$ and from 0.66 to 1.65$\varnothing$, with an average of 0.88$\varnothing$, for the Agios Stefanos section.

The mean graphical values of kurtosis in the Kefalonia Island indicate the predominance of the leptokurtic field (mean 1.01, 1.03 and 1.03$\varnothing$) for the Koutala beach, Xi beach and Matzavinata section, respectively and the predominace of the platykurtic field (mean 0.82$\varnothing$) for the Chavdata section. The mean graphical values of kurtosis of the Corfu Island indicate the predominance of the leptokurtic field (average 1.17$\varnothing$) for Arillas section and the platykurtic field (average 0.88$\varnothing$) for Agios Stefanos section.

Peakedness is dominated by platykurtic behavior with more than half of the samples documenting the same tendency. It is then followed by mesokurtic and leptokurtic behaviors in equal measure and very platykurtic and very leptokurtic behaviors. Changes in kurtosis values are attributed to variations in the flow characteristics of the deposition medium [79]. The predominance of leptokurtic $K_G$ suggests the accumulation of fine particles in a primarily low-energy marine environment [80].

## 7. Discussion

### 7.1. TOC and CaCO₃ Correlations

In the Kefalonia samples, a series of alternations of positive and negative correlation of the two parameters was observed. In any case, the low percentages of $CaCO_3$ probably indicate low rates of sedimentation (probably linked to the gradual filling of the basin), while you observe a progressive decrease in the percentage of TOC, possibly linked to a decrease in organic productivity. The frequent changes in the correlation of these two parameters (discontinuation of this relationship within the stratigraphic column) indicates that $CaCO_3$ in the sediments is mainly controlled by its dissolution in the subsurface water layer and sediments due to the decay of organic material.

In the section of Agios Stefanos, only positive correlation of the two parameters is observed, with minimal negative correlations, while in the section of Arillas seem to have alternating negative and positive correlations. Low rates of $CaCO_3$ probably indicate low rates of sedimentation, while higher rates of TOC are probably associated with more organic productivity. The frequent changes in the correlation of these two parameters (discontinuation of this relationship within the stratigraphic column) indicates that $CaCO_3$ in the sediments is mainly controlled by its dissolution in the subsurface water layer and sediments due to the decay of organic material. The decrease in the percentage of $CaCO_3$ from cycle 1 to the following cycles indicates at the same time the gradual filling of the basin, while it can also be linked to the cold climatic conditions that prevailed at the end of the Miocene in the study area [81], where they probably led to an increase in $CaCO_3$ solubility.

The study of both TOC and $CaCO_3$ content and the identification of correlations between the two parameters (Figure 13) can provide several clues about the depositional environment and conditions. When comparing the two islands, it is evident that there are differences in the distribution of $CaCO_3$. The sediments found in Kefalonia exhibit lower percentages of $CaCO_3$ compared to those in Corfu, indicating that the former was deposited in a shallower sedimentation environment. Additionally, the concentration of $CaCO_3$ in Kefalonia reveals more stable conditions in the sedimentation cycles in contrast to Corfu. The sediments in Corfu appear to have been deposited in an increasingly shallow environment due to the gradual filling of the basin, while those in Kefalonia indicate that the depth of the basin remained constant during sedimentation. This comes in agreement with the biostratigraphic results as the depositional environment of Kefalonia is indicated as a deep-water setting while Corfu is characterized by deep-water setting evolving to a shallow lagoon system of limited connectivity to the sea. Furthermore, the distribution of TOC also shows differences between the two islands. The sediments in Kefalonia display higher TOC percentages than those in Corfu. Moreover, there is a noticeable difference in the concentration of TOC in Kefalonia, where the content of TOC samples decreases upward, compared to Corfu where an upward increase in TOC samples is observed.

The periodic variation of the percentage of TOC in a permanently submerged basin is closely related to tides, water current, and sediment deposition and disturbance processes. This environment is further supported by the conducted biostratigraphic analysis, documenting frequent fluctuations in depth and surface water temperature and salinity. Tides affect water circulation and the transport of particles, including organic material, thereby affecting their distribution and deposition in the basin. The recycling of organic material can be affected by water circulation and particle distribution. Organic material can undergo various processes of breakdown and decomposition that depend on the conditions of the basin. This can be explained as during the tidal cycles (period of high and low tide), the

direction and speed of the water changes. Increased water velocity during high tide can affect bottom disturbance and redeposition of sediment, including organic material. This process brings the covered organic material into a more oxidizing environment resulting in its partial preservation. In the opposite case, during low tide, when the water velocity decreases, organic particles tend to be deposited.

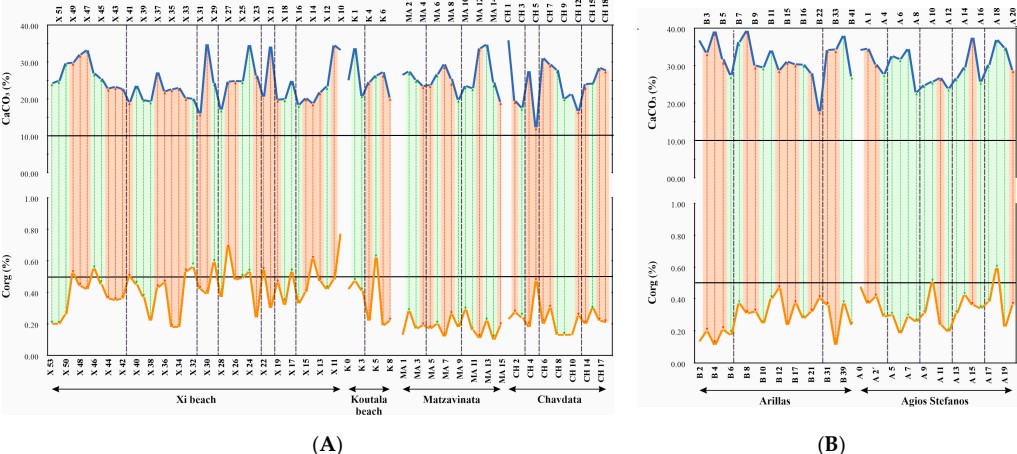

(**A**)  (**B**)

**Figure 13.** Correlation of $CaCO_3$ (blue line) and TOC (orange line) contents within studied deposits. The green dashed line illustrates positive correlation, while the red dashed line illustrates negative correlation. The black lines illustrate the recommended value (10%) for optimal plasticity and (0.5%) for sediments rich in organic content, respectively. The dashed black lines illustrate the mentioned cycles in sedimentation. (**A**) Kefalonia Island samples; (**B**) Corfu Island samples.

### 7.2. Extensive XRF Analysis and Material Utilization Recommendations

The analysis encompasses XRPD analysis, pH determination, plasticity determination, specific surface area and thermal analysis and a comprehensive discussion on this can be found in [17]. XRPD analysis highlights variations in minerals across different locations. pH measurements, crucial for assessing material suitability, range from 8.1 to 9.6, indicating that the sediments are suitable for pelotherapy use. Plasticity tests categorize most samples as low plasticity, with limits ranging from 17 to 48% (PL), suggesting potential benefits. Specific surface area values slightly below limits indicate room for improvement, and the correlation between these factors and material quality suggests that maturation with thermometallic water could enhance therapeutic clay properties.

### 7.3. Interrelationship of Textural Parameters

To differentiate between various depositional environments, a combination of multiple textural parameters is used in the form of bivariant plots [82]. The underlying concept of these plots is rooted in the belief that statistical parameters can effectively represent changes in the fluid flow mechanisms governing sediment transportation and deposition [83]. Numerous researchers have proposed that these plots can serve as a dependable tool for distinguishing between the processes associated with different sedimentation environments [83–87].

#### 7.3.1. Bivariate Plot between Mean Size and Sorting

The bivariate plot of sorting ($\sigma_1$) versus mean ($M_z$) (Figure 14Aa,b,Ba,b) reveals that the samples exhibit poor to very poor sorting and are primarily composed of silt and clay. Sediments that are well-separated during prolonged transport tend to have relatively uniform grain sizes. If different sizes are present, it suggests that the sediment was deposited rapidly [88]. The samples consist predominantly of clay and are very poorly sorted, indicating that they spent a considerable amount of time in the transport and deposition medium. The bivariate plot of standard deviation ($\sigma_1$) versus mean ($M_z$)

shows that the samples cluster near the extreme right-hand end of the standard inverted V-shaped trend described by Folk and Ward [58]. This clustering implies a narrower range of grain sizes.

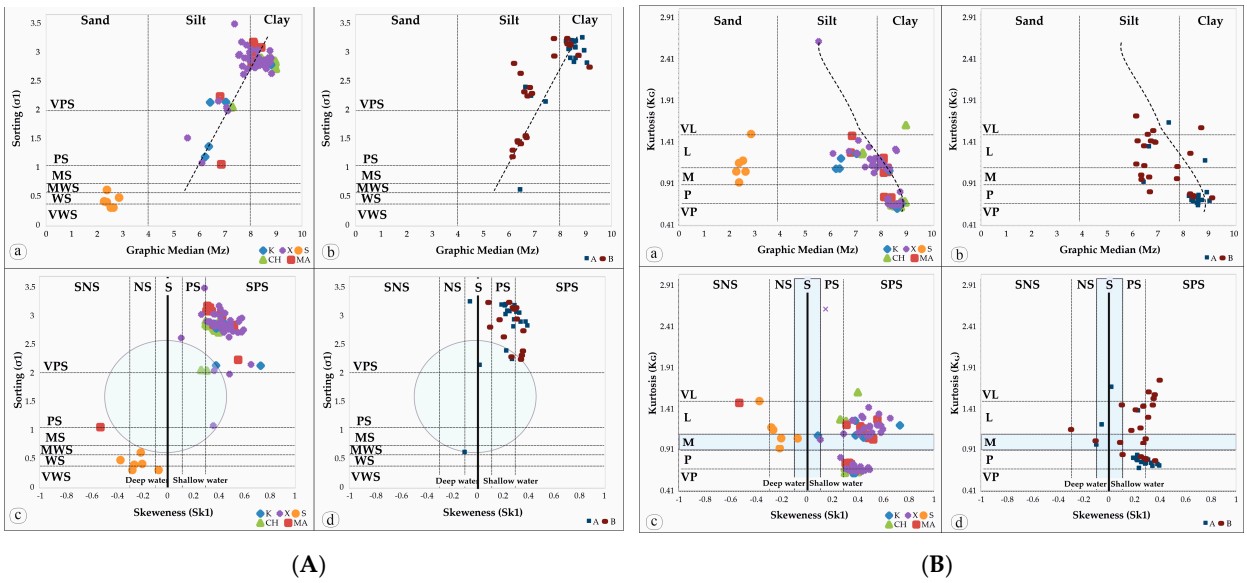

**Figure 14.** Bivariate plots of samples in the model plot as proposed by [58]. (**A**) (**a**,**b**) mean grain size vs. standard deviation, (**c**,**d**) standard deviation vs. skewness. Graphs "a and c" refer to Kefalonia Island samples while "b and d" refer to Corfu Island samples; (**B**) (**a**,**b**) mean grain size vs. kurtosis, (**c**,**d**) kurtosis vs. skewness. Graphs "a and c" refer to Kefalonia Island samples while "b and d" refer to Corfu Island samples.

The sediments are generally monomodal, with clay being the dominant component. Both mean grain size and sorting are primarily influenced by hydraulic processes, such as suspension settling [89]. However, there is an exception with the "S" samples, which correspond to the sand collected from the Xi beach section. These samples are fine sand and well sorted, indicating that they have spent a reasonable amount of time in the transport medium and have undergone grain-size reduction due to grain-to-grain interaction. Their classification as well- to moderately well-sorted sediments suggest continuous reworking by currents and waves [90]. The bivariate plot of standard deviation ($\sigma 1$) versus mean ($Mz$) for the sand samples also shows clustering near the extreme base of the standard inverted V-shaped trend described by Folk and Ward [58], indicating a narrower range of sand grain sizes. The nature of the sediments remains monomodal, with fine-grained sand as the dominant component. Both the average grain size and sorting in these sand samples are primarily hydraulically controlled [89].

### 7.3.2. Bivariate Plot between Skewness and Standard Deviation

The bivariate plot of the standard deviation ($\sigma_1$) against the skewness plot ($Sk_1$) (Figure 14Ac,d,Bc,d) reveals that most samples are characterized as very poorly sorted and exhibit positive to very positive skewness. The samples tend to cluster outside the circular ring, as suggested by Folk and Ward [58]. This pattern, along with the generally positive to very positive symmetry ($SK_1$ around 0.3), suggests that the samples are generally bimodal, with poor to very poor sorting.7.3.3. Bivariate Plot between Mean Size and Kurtosis

The bivariate plot of kurtosis ($SK_1$) versus mean ($M_z$) (Figure 15Aa,b,Ba,b) demonstrates that most samples exhibit positive to very positive skewness, with only five samples showing symmetry. Two distinct groups of values are observed, aligning with the standard trend curve proposed by Folk and Ward [58]. One group of samples clusters near the inner part of the sine curve, while another group falls on the right edge of the sine curve, indicating two distinct grain size ranges. This sinusoidal pattern in the sediments is a

result of a proportional mixture of two sediment size classes, specifically clay and silt. This pattern is consistent with the inverted V-shaped curve as suggested by Folk and Ward [58]. Generally, unimodal sediments tend to be nearly symmetrical, but the mixing of sediment size classes leads to positive skewness due to an excess of clay in the size class proportions within the mixture.

The plot of mean size ($M_Z$) against kurtosis ($K_G$) reveals the influence of mixing two or more size classes of sediment, which affects the sorting of the central and tail portions of the curve. According to Folk and Ward [58], the presence of only one mode results in a nearly normal curve ($K_G = 1.0$). The addition of a small amount (3–10%) of another mode results in poorer sorting in the tail and better sorting in the center, making the curve very leptokurtic with $K_G > 1.0$. When two modes are present in roughly equal amounts (ranging from 25–75% to 75–25%), a very platykurtic behavior is observed [91]. Most of the studied samples exhibit very platykurtic to platykurtic behavior, with three samples showing mesokurtic behavior and another three showing very leptokurtic behavior with a constant presence of coarser grain. According to [92], this indicates that most of the studied samples are dominated by silt, and in the case of bimodal sediments, the second mode is present in a subordinate amount.

### 7.3.3. Bivariate Plot between Skewness and Kurtosis

The bivariate plot of kurtosis ($K_G$) versus kurtosis ($SK_1$) of a given sediment population is a powerful tool for distinguishing between depositional environments, as emphasized by Friedman [93]. The samples (Figure 15Ac,d,Ba,b) are predominantly leptokurtic to mesokurtic, with the majority falling into the leptokurtic range. Additionally, most of the samples exhibit positive to very positive skewness. According to Friedman [94], extreme values of kurtosis, whether high or low, suggest that some of the sediment had undergone sorting in a high-energy environment distinct from the current depositional setting. It is worth noting that the plot of kurtosis (KG) versus asymmetry ($SK_1$) does not follow the typical sinusoidal path as mean size changes. Instead, it depends on the presence of two modes within the sediment, as suggested by Folk and Ward [58].

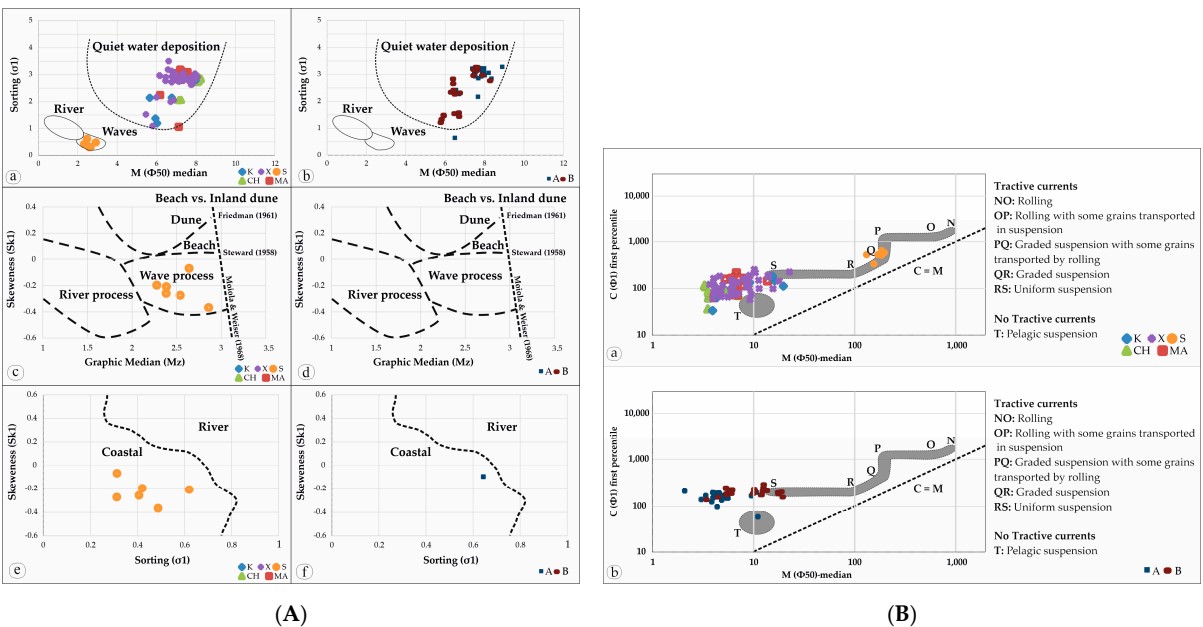

**Figure 15.** (**A**) Bivariate plots of various parameters. (**a,b**) median vs. standard deviation (after [90,95]), (**c,d**) mean size vs. skewness (after [90,95]) and (**e,f**) skewness vs. standard deviation (after [94]). Graphs "a, c and e" refer to Corfu Island samples while "b, d and f" refer to Kefalonia Island samples; (**B**) CM diagram [96] depicting depositional processes for the (**a**) Kefalonia Island samples and (**b**) Corfu Island samples.

*7.4. Bivariate Grain-Size Parameters*

Statistical parameters obtained through graphical methods were utilized to create various bivariate diagrams and gain insights into the prevailing environmental conditions at the time of deposition.

Friedman [94], and Moiola and Weiser [90], plotted mean grain size vs. skewness (Figure 15Aa,b) for differentiating river, beach, and coastal dune sub-environments. This method works effectively in separating beach from river sands, as well as river from coastal dune sands, and the differentiation remains consistent whether using quarter, half, or whole phi data [90]. The graphical data do not display samples in this diagram; however, it is noted that the Xi sand samples are categorized as beach sub-environment.

Stewart [95] plotted mean grain size vs. sorting to differentiate between river, wave, and quiet water processes. The plots (Figure 15Ac,d) from graphical analyses indicate that all the samples were deposited under quiet water conditions, with particles in suspension. It is noted that the Xi sand samples were deposited under wave influence.

Sorting vs. skewness [90,94] was plotted to distinguish beach and river sub-environment. The graphical data do not plot the samples (Figure 15Ad,e), except for the Xi sand samples, which are categorized under the beach sub-environment.

Passega [96] introduced the utilization of the C-M plot as a tool to scrutinize the hydrodynamic forces that governed sediment deposition. This diagram (Figure 15B) was constructed by juxtaposing two critical metrics: the coarsest grain in the sample, referred to as C (expressed in microns), and M, which signifies the median grain size, also measured in microns, on a logarithmic probability curve. In general, the grain size of clastic sediment serves as a defining parameter for characterizing the hydraulic energy state of the environment [92,97,98]. The analysis of the sampled materials plotted on the C-M diagram distinctly reveals that these samples underwent transportation via suspension, which serves as a compelling indicator of the prevailing low-energy conditions.

*7.5. Linear Discriminate Analysis (LDF)*

Linear discriminant function (LDF) analysis plays a crucial role in the identification of the depositional environment at the time of sedimentation. According to Sahu [59], there exists a distinct correlation between variations in energy and fluidity and the diverse processes and settings where sediment accumulates. In the present investigation, both linear and multi-group discriminant function analyses [59,99] were applied to further discriminate between depositional environments.

Upon computing the LDF (Y1, Y2, Y3, and Y4) for the samples from Kefalonia, it was ascertained that the values of Y1, Y2, Y3, and Y4 span a range from −14.54 and 21.94, 63.33 and 947.33, −106.72 and 1.26, and 5.10 and 18.00, respectively. Similarly, the LDF (Y1, Y2, Y3, and Y4) for the samples from Corfu, encompass values between −12.66 to 14.60, 144.10 to 868.38, −92.12 to −1.27, and 7.03 to 15.65, respectively. Regarding Y1 values (Figure 17Aa,b), most of the samples are indicative of deposition in a shallow agitated water environment, while a smaller subset suggests a beach environment. In terms of Y2 values (Figure 17Ac,d), all the samples are characteristic of sedimentation in a shallow marine environment. Furthermore, as per the Y3 values (Figure 17Ba,b), all the samples are associated with the shallow marine environment, except for one sample from Agios Stefanos, which is linked to a deltaic or lacustrine setting. In the case of Y4 values (Figure 17Bc,d), some samples align with the density current deposits field, while others align with deltaic deposits.

The binary plot of Y2 versus Y1 (Figure 16Aa,b), following the approach proposed by Sahu [59], reveals that most samples are clustered within the shallow disturbed water/shallow marine environment, with only a few samples being associated with the beach/shallow marine environment. Additionally, the plotting of Y3 versus Y2 and Y4 (Figure 16Ac,d,e,f) indicates that all the samples were deposited in shallow marine environments. The plotting of Y4 versus Y3, Y2, and Y1 (Figure 16B), respectively, demonstrates that

that some samples are positioned within the density current deposits field, while others are aligned with deltaic deposits.

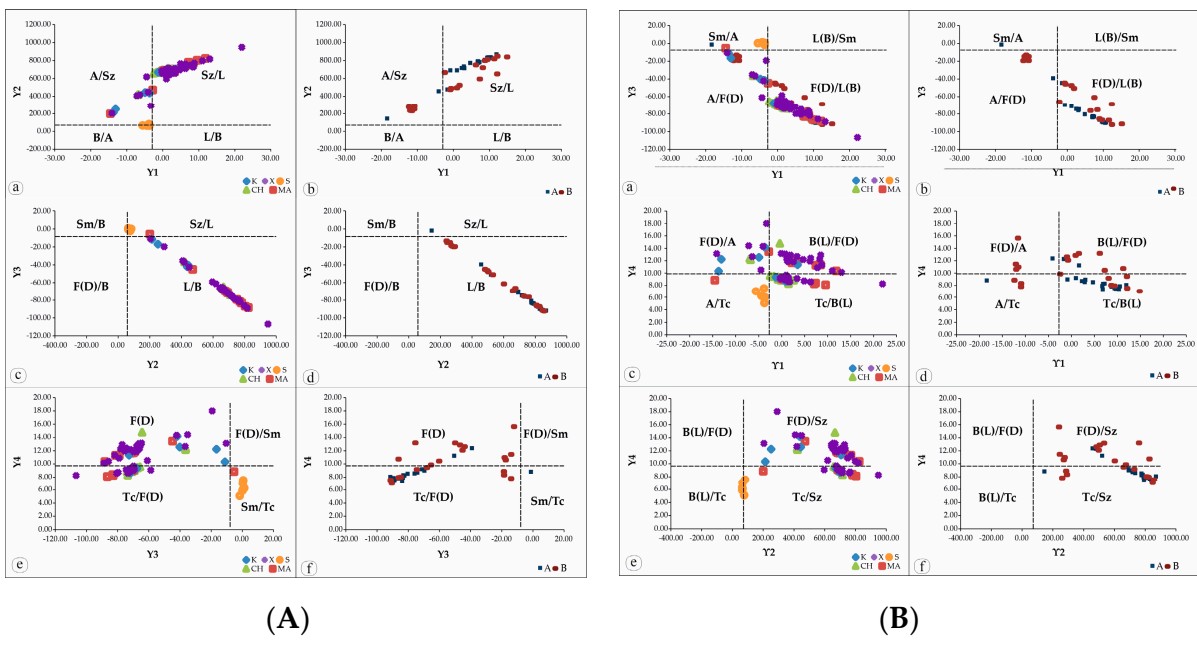

**Figure 16.** (**A**) Linear discriminate function analysis plot for (**a**,**b**) Y 1 vs. Y 2 discriminates between beach and aeolian environment; (**c**,**d**) Y 2 and Y 3 between beach and shallo0.w marine sub-environment; and (**e**,**f**) Y 3 vs. Y 4 discriminates between marine turbidity and fluvial environment. Graphs "a, c and e" refer to Kefalonia Island samples, while "b, d and f" refer to Corfu Island samples; (**B**) Linear discriminate function analysis plot for (**a**,**b**) Y 1 vs. Y 3 discriminates between beach and aeolian environment; (**c**,**d**) Y 1 and Y 4 between turbidity and fluvial environment; and (**e**,**f**) Y 2 vs. Y 4 discriminates between marine turbidity and fluvial environment. Graphs "a, c and e" refer to Kefalonia Island samples, while "b, d and f" refer to Corfu Island samples. Abbravations: A = Aeolian, Sz = Surf zone, L= Littoral, B = Beach, F(D) = Fluvial (Deltaic), Sm = Shallow marine, Tc = Turbidity currents.

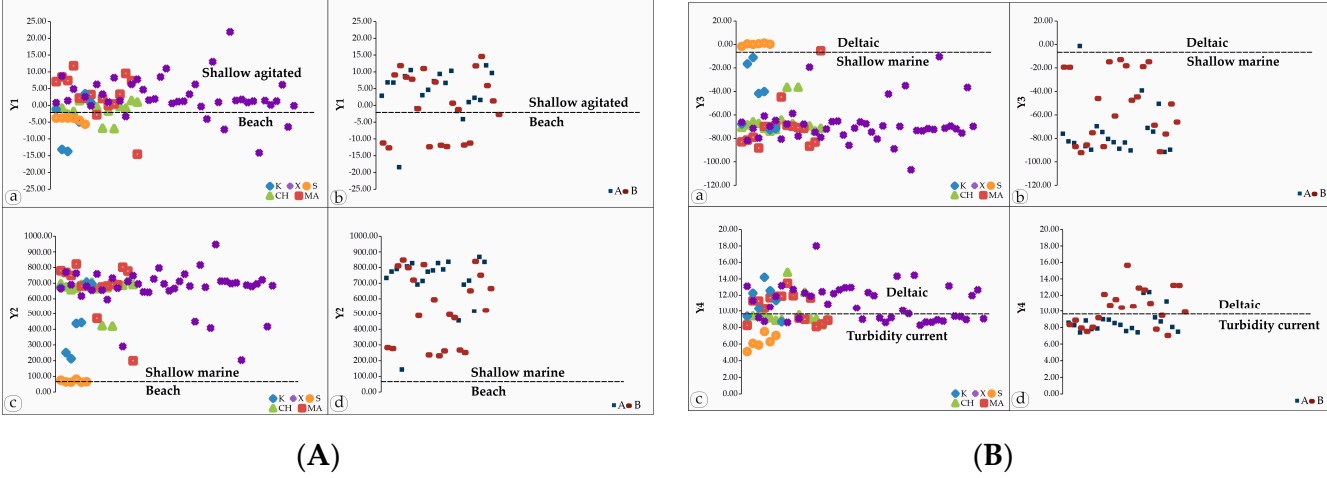

**Figure 17.** (**A**) Linear discriminate function analysis plot for Y 1 which discriminated shallow agitated from beach environment and Y 2 discriminating beach and shallow marine environment. Graphs "**a** and **c**" refer to Kefalonia Island samples, while "**b** and **d**" refer to Corfu Island samples; (**B**) Linear discriminate function analysis plot for Y 3 which discriminated shallow marine from deltaic environment and Y 4 discriminating turbidities and deltaic environment. Graphs "**a** and **c**" refer to Kefalonia Island samples, while "**b** and **d**" refer to Corfu Island samples.

As noted previously, LDF can classify the samples into five distinct depositional environments: dune, river, beach, shallow marine, and turbidity. Upon plotting the studied samples on the diagrams, the overarching conclusion is that all the samples unambiguously cluster within the shallow marine environment and turbidity. This outcome signifies that these sediments were initially deposited by suspension and gravity flow deposits, subsequently subjected to sorting processes driven in a shallow marine environment affected by tidal action.

### 7.6. Parameters and Applications in the Environments of Deposition at Kefalonia and Corfu Islands

The application of statistical techniques holds potential for elucidating the attributes of the depositional setting. Nonetheless, a comprehensive study requires the consideration of alternative methodologies, encompassing a spectrum of data such as the grain size, sedimentary structures, bedding dimensions, and sedimentary architecture.

In the Pliocene of Kefalonia Island, there is short stratigraphic hiatus and transgressive well-bedded conglomeratic facies, overlain by limestone beds passing upwards into sandstone, and sandy limestone with layers of blue marls. This lower cycle, which is also characterized by the presence of gypsum, is followed by a cycle composed of blue marls, clays, and marly limestone, occasionally interbedded with sand and an upper cycle consisting of limestone and sand. Up-section, there are cyan-colored clays rarely interbedded with sandstone, less than 10 cm thick, predominantly at Xi Beach and at Koutalas Beach, a limestone bed is observed, transitioning upwards to yellowish sandstone, and marly limestone, occasionally interbedded with thin marls. Considering the Sahu [59] parameters (Figure 18), this basin seems to follow a complex deepening and shallowing history. These features, along with the presence of muddy debris flow in the section of Xi beach ascribe to a shallow marine environment, influenced by slow sedimentation through suspension settling and sediment-density flows. Furthermore, the presence of lenticular bedding (Figure 5Bb) is a distinctive sedimentary structure characterized by alternating layers of mud and sand. It is typically formed during periods of reduced water flow or slack water conditions when suspended mud particles settle on top of small sand formations as the water's velocity decreases to near-zero levels. This evidence is also confirmed by the statistical parameters described above, further supporting that the lenticular beds are commonly encountered in high-energy environments, particularly in the intertidal and supratidal zones [100], documenting evidence of tidal rhythms, tidal currents, and slack water conditions.

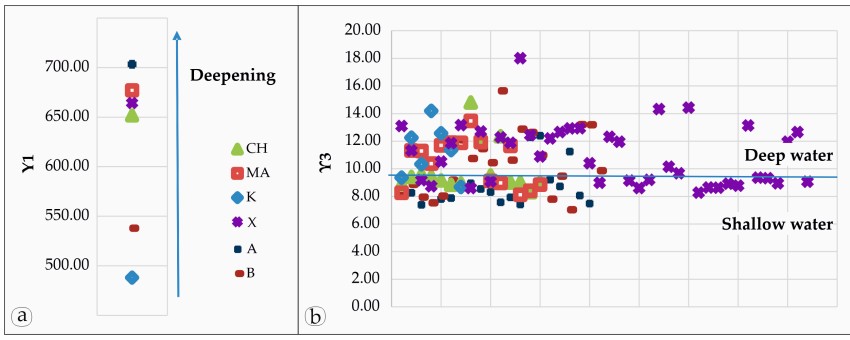

**Figure 18.** Coefficient of discrimination between beach and shallow sea environments with (**a**) average value of Y 1 per studied section and (**b**) Y 3 for each sample of the studied areas.

Similarly, along the coastal area of Arillas, at the lowermost section of the stratigraphic sequence, there is a gradual transition from flaser to wavy, and further to lenticular bedding. Furthermore, up-section, the sediments of Agios Stefanos section, are marked by the prevalence of gray silt, extending up to 1 m, occasionally interbedded with sandstone lenses (size of mm to few cm), passing into silt-sand alternations that exhibit gradually diminishing thickness in the sandstone layers. Sandstone layers >10 cm show silt and/or

rarely very thin organic-rich layers and/or silt-sand alternations with hummocky cross-bedding in the sandstone layers, indicating a shallow marine environment affected by storms. The prevalent occurrence of lenticular, wavy, and flaser bedding, along with the presence of mud drapes (Figure 10f) in Agios Stefanos, indicates a pattern of alternating conditions involving both flowing and standing water. These sedimentary structures suggest a depositional environment characterized by dominant flowing currents and periodic reversals [101–104]. When considering the marine trace fossil assemblage in conjunction with these features, it implies that the sandstones were likely deposited in an environment influenced by tides, further explaining the bimodal character of the sediments reported by the statistical parameters.

In the context of FTBs and geologically active basins, coastal environments typically display low energy attributes, with tidal forces exerting a dominant influence, particularly in deltaic zones and estuarine systems [105–108]. Furthermore, it has been established that deltaic and estuarine flood events can generate density currents closely resembling turbidity currents [109–112]. A study of lithofacies evolution reveals significant changes in energy levels and an overall deepening of the basin over time.

### 7.7. Physical–Chemical Characteristics and Applications in Therapectical Mud

Unconsolidated sediments and sedimentary rocks are comprised of individual grains, and their attributes and classifications are primarily based on texture and chemical–mineralogical composition [113]. $CaCO_3$ contents greater than 10% usually negatively affect the plasticity of the clay [114]. Therefore, the average 26.71% (Table 1 and Supplementary Materials Table S1) $CaCO_3$ content of the studied samples, which is significantly higher, is theoretically a negative quality characteristic. However, it does not seem to affect the mechanical behavior and plasticity of the clay and is therefore a neutral quality criterion. The lack of effect on plasticity is probably because $CaCO_3$ was precipitated along sediment of very low grain size (Tables 3 and 4), which, if not favorable, at least does not adversely affect the mechanical properties of mud.

Concentrations of organic matter ranging from 2–5% accelerate the maturation of clay, facilitate the formation of sulfur compounds, which exhibit vasodynamic behavior and therapeutic action in various diseases, increase the cation exchange capacity (C.E.C.) of the clay, and enhance its plasticity [115–117]. Concentrations of organic matter, mainly peat, exceeding 10%, usually do not disperse in water, resulting in a reduction in the clay plasticity and a general deterioration of its mechanical behavior. The TOC content (<0.77%) (Table 2 and Supplementary Materials Table S1) is much lower (below the threshold of 2%), for achieving the maturation of materials for use in mud therapy, therefore, the enrichment of samples with organic matter will be required. On the other hand, the TOC percentage favors its mechanical properties, as concentrations greater than 10% reduce the plasticity of the mud and generally degrade its mechanical behavior [115–117].

When assessing sediments in terms of grain size, they can be categorized according to a size scale, with the most widely adopted classification being the one originally proposed by Wentworth [118]. This classification divides the sediment's clasts into six primary classes based on their size. Typically, sediment consists of one or more size classes, which is why size classification should account for the relative proportions of each constituent class. This is important because various essential properties are directly or indirectly linked to the grain size of the sediment. These properties include air and water permeability, plasticity (and, by extension, workability), water retention capacity, and nutrient availability.

The grain size classes (Figure 19) to which they belong are the following:

- "Silt": With a prevalent silt component, represented on average by 73.86% of the total sample, followed by clay values with an average of 21.94% and those of sand (4.20%), which is the minor component of the sample.
- "Mud": With a prevalent silt component, represented on average by 53.52% of the total sample, followed by the values of clay with an average of 42.57% and those of sand (3.91%), which constitutes the minor component of the sample.

- "Sandy Silt": With a prevalent silty component, represented on average by 65.42% of the total sample, followed by the values of clay with an average of 20.61% and those of sand (14.06%), which constitutes the minor component of the sample.
- "Sandy Mud": With a prevalent silt component, represented on average by 43.60% of the total sample, followed by silt values with an average of 35.43% and those of sand (20.97%), which is the minor component of the sample.

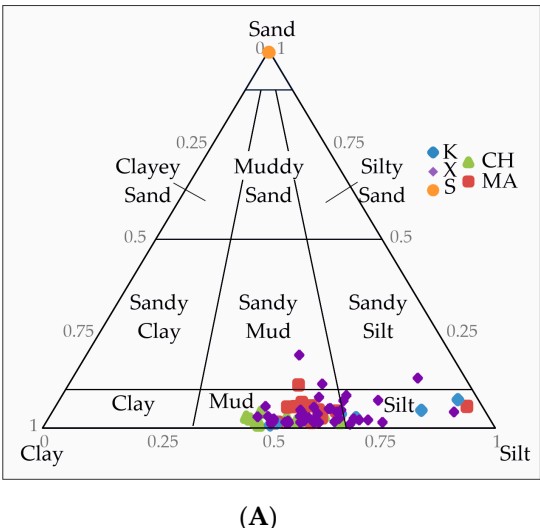

(**A**)

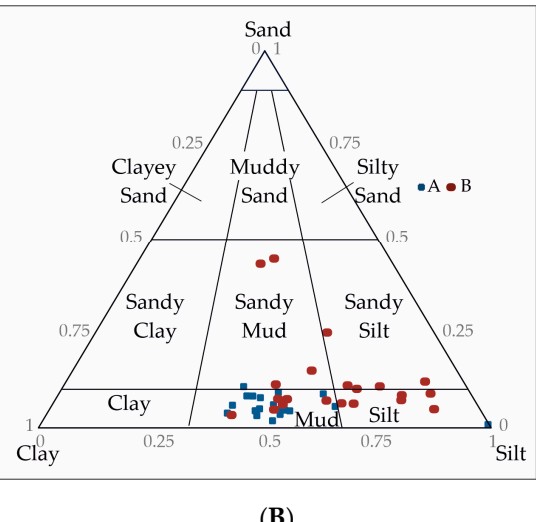

(**B**)

**Figure 19.** Grain size ternary diagram after Folk and Ward [58], for classification of the sediments from the different studied sections: (**A**) Kefalonia Island and (**B**) Corfu Island.

"Silt" represents most of the samples, represented by 73.45% of the samples in total, followed by "Mud" (14.16% of the samples), "Sandy Mud" and "Sandy Silt" (7.08% and 5.31%, respectively).

Grain size analysis offers several important considerations, with the first being the assessment of its suitability for skincare products. Ideally, a product should have the lowest possible sand content since coarser fractions can potentially cause skin irritations or injuries [119] (and references therein). Furthermore, a higher clay content is generally preferred over a silty component. This is because the clay fraction, when present in the same volume, offers a greater specific surface area, increasing the reactivity of the mud [119] (and references therein). For instance, it enhances cation exchange capacity [120] (and references therein). Based on the conducted statistical analysis, shallow sea sedimentary environments exhibit greater homogeneity in terms of the clay/silt ratio compared to tidal environments, as this environment favors the bi-modal character of grain sedimentation. As a result, the former, used in mud therapy, requires more processing due to a higher sand content.

Among all the samples analyzed, from a grain-size perspective, the "optimized" area for the potential development of clay-based cosmetic products is Chavdata in Kefalonia. This area is classified as "silt" and contains only 0.57–3.94% sand, with an average of 1.85%. Furthermore, considering that the "sandy-mud" and "sandy-silt" samples make up just 12.39% of the total, and the average sandy component is below 10%, all the study areas offer potential for the development of clay-based cosmetic products. Moreover, the sampled areas' locations, especially Xi Beach in Kefalonia and Agios Stefanos in Corfu, are well-known to tourists for their natural mud, which is ideal for exfoliating the skin and treating issues such as insect bites and dry skin conditions.

Additionally, it is essential to consider the grain-size distribution (by weight) of the sandy fraction. According to [121], the comparison of Makirina Bay peloid mud with other raw materials that are successfully used in cosmetics and/or for therapeutic purposes shows that Makirina Bay peloid mud has comparable examined properties. The

sedimentological analysis indicated that it mainly consists of sandy silt, poorly sorted and the depositional environment was relatively calm. Even though the reported presence of quartz, organic debris, and broken bioclasts can increase the sand-sized particles in peloid mud, the Makirina Bay peloid mud was considered to not be problematic since most of the sand particles are fine sand-sized (<250 μm). It was also proposed that sand-sized particles can be separated by sieving before using peloid mud. Notably, this study samples dominant contribution is silt, which are finer compared to Makirina Bay peloid mud samples. Even in the case of sandy fraction, the primary grain-size is very fine sand. Furthermore, the presence of very coarse, coarse, medium, and fine sand components is absent. Even when examining the frequency curves, the fraction below 3∅ constitutes less than 5%. Therefore, it has been confirmed that the presence of the sandy fraction in mud, following its application, should not cause irritation or abrasive issues. According to the Udden–Wentworth classification [118], the sand dimensions corresponding to "fine sand" and "very fine sand" fall within the grain size range of 125–63 μm. This aligns with other studies [119,121] on the grain size of natural mud, which also supports the notion that these muds have a superior quality and are gentler when applied to the skin.

## 8. Conclusions

The study offers valuable insights into optimizing mud selection as a cosmetic resource with positive effects on human health and well-being. The investigation involves a comprehensive analysis, incorporating $CaCO_3$, TOC, grain size, and statistical parameters, across six outcrops on the Kefalonia and Corfu islands.

- Although there is a slight disparity in the average $CaCO_3$ content between Corfu island samples (30.90%) and Kefalonia island samples (24.60%), both percentages significantly surpass the recommended 10% value for optimal plasticity. However, no observable impact on the mechanical behavior and plasticity of the clay was noted, rendering it a neutral quality criterion.
- Corfu island samples exhibit higher TOC content, suggesting potential superiority for mud therapy. Nevertheless, all samples show a TOC content (<0.77%) considerably below the required threshold (2–5%) for material maturation in mud therapy. Therefore, enriching the samples with organic matter is required.
- Although TOC values are below the recommended maturity level, the TOC content (<0.77%) on both islands samples does not affect the mechanical behavior of the mud (TOC > 10% has a negative effect), indicating its potential for therapeutic purposes.
- Cumulative frequency percentage curves and grain-size statistics classify the studied samples as fine- to very fine-grained sediments (clay to very fine sand).
- Linear and multigroup discriminant analyses categorize two sediment types: a unimodal type, characterized by mud grain-size dominance, deposited in shallow water environments, and a bimodal type, featuring mud and sand content alterations, deposited in tidal-affected environments.
- Based on the statistical analysis, sedimentary environments in shallow seas exhibit more uniformity in the clay/silt ratio compared to tidal environments. This uniformity, characteristic of the samples from Kefalonia island, is preferable for mud therapy utilization.
- The studied sediments from Corfu island also have potential for mud therapy but require additional processing due to elevated sand content.

**Supplementary Materials:** The following supporting information can be downloaded at: https://www.mdpi.com/article/10.3390/geosciences14020048/s1, Figure S1: Xi stratigraphic column; Figure S2: Koutala stratigraphic column; Figure S3: Matzavinata stratigraphic column; Figure S4: Chavdata stratigraphic column; Figure S5: Arillas north stratigraphic column; Figure S6: Arillas south stratigraphic column; Figure S7: Agios Stefanos stratigraphic column; Figure S8: Cumulative curves of Kefalonia samples; Figure S9: Cumulative curves of Corfu samples; Table S1: Statistical parameters; Table S2: $CaCO_3$ and TOC percentages.

**Author Contributions:** Conceptualization, C.B., G.I. and A.Z.; methodology, C.B., P.P., G.I. and A.Z.; software, C.B.; validation, C.B., P.P., N.D., A.K., G.I. and A.Z.; formal analysis, C.B., G.I. and A.Z.; investigation, C.B., N.B., E.Z., P.P., N.D., A.K., P.Z., D.C.A., G.I. and A.Z.; resources, G.I. and A.Z.; data curation, C.B., P.P., G.I. and A.Z.; writing—original draft preparation, C.B.; writing—review and editing, C.B., P.P., G.I. and A.Z.; visualization, C.B., G.I. and A.Z.; supervision, C.B., G.I. and A.Z.; project administration, G.I. and A.Z.; funding acquisition, G.I. and A.Z. All authors have read and agreed to the published version of the manuscript.

**Funding:** This research was funded by Project "IONIAN ISLANDS 2014–2020" ACT: Recording and presentation of Geosites & Georoutes of the Geopark Kefalonia—Ithaca with the aim of joining of UNESCO Geoparks, Sub-project 2 "Recording and study of Geodiversity & Biodiver-sity of the Kefalonia—Ithaca Geopark" PROJECT CODE (FK/MIS): 5007956/81155.

**Data Availability Statement:** Data available as Supplementary Materials.

**Acknowledgments:** We would like to thank Tzesi Zanai at University of Patras, department of Geology for her valuable contributions to our field work and microscopic analysis, and the reviewers for their constructive comments that increased the manuscript quality.

**Conflicts of Interest:** The authors declare no conflicts of interest.

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
