# Peer review of "The Knowledge and Application of Sedimentary Conditions of Shallow Marine and Tidal Waters of Ionian Islands, Greece: Implications for Therapeutic Use"

_geosciences, doi:10.3390/geosciences14020048_

Round 1
Reviewer 1 Report
Comments and Suggestions for Authors
Review of:
‘’The knowledge and application of sedimentary conditions of 2 shallow marine and tidal waters of Ionian Islands, Greece: Implications for therapeutic use’’
Good paper. Some concern about the length and the large number of references.
Specific comments
-Introduction line 49: consider changing the word ‘’conditions’’ to ‘’ailments’’.
- Final paragraph of Introduction: there are some issues in the entire paragraph. For example, in line 71 the phrase ‘’and we tried to find equivalent areas to protect the existing coasts’’ is not very clear in its meaning. There are some repetitions and the aim of the study is mentioned in the beginning and end of the paragraph separated by some bibliographic references. Finally, there is mention of a future paper and it is not clear what is presented in the present study and what issues may remain as questions for another publication (usually further goals that present themselves as outcomes of a study or suggestions for future work are usually mentioned in the conclusions section). I suggest that the authors amend the entire last paragraph of the introduction.
-Materials and methods: not very clear number of samples per study section( i.e. Xi beach 200 mudstone and 5 sand and from those 20 selected were subjected to micropaleontological analysis, and 43 to geochemical ?? but in table S1 48 samples of CHI have grain size analysis results). Perhaps the authors should add a supplementary table showing the number of samples per section and how many were subjected to which further analysis or improve the sample description in the corresponding materials and methods paragraph.
-The authors should consider amending the general title ‘’Results’’ of section 6 making it more detailed because sections 4 and 5 also present other types of results.
-In the calcium carbonate results the authors should verify that the presentation of numerical values with as many significant digits is appropriate i.e. 35.72% instead of 35.7%. The numerical results are usually rounded to the appropriate significant digits according to analytical method uncertainty (i.e. a value of 10 measured with 1% uncertainty should be presented as 10.0 and not 10.00). The same consideration should also be applied to the presentation of grain size results in section 6.3 unless it is common practice for these results to be presented with so many significant digits. Also see the percentages in section 7.6 (lines 964-965).
-Line 428: not clear what MT is. In figures 11 and 12 the legends contain the initials MT, MC, MK but it is not clear what these values are.
-Section 6.4: it is not very clear what the result here is. Probably the last sentence should be elaborated. (as I understand cumulative frequency curves were plotted but are not presented, they were characterized by moderate gradients therefore the result is lack of sorting and low kinetic energy and velocity environment?)

Good quality
Reviewer 2 Report
Comments and Suggestions for Authors
The article “The knowledge and application of sedimentary conditions of shallow marine and tidal waters of Ionian Islands, Greece: Implications for therapeutic use” is devoted to the analysis of the geological composition of tidal environments, which formed fine-grained clay and mud sediments at the Ionian Islands. As the authors argue, these sediments possess unique therapeutic properties. Unfortunately, there is no data about the chemical and mineralogical compositions of these clay and mud. The title of the manuscript does not match the content. In the paper, the detailed geological processes of marine deposit forming, the impact of dynamics of the tidal process on the carbonate and organic mud sedimentation, and their grain-size composition are considered. However, there is no information about the geochemical compositions of these sediments, their characteristics, and the criteria for application in medicine and cosmetology. The authors should change the title and make another accent or give the detailed characteristics (chemical, mineralogical composition, the medical effect from application, etc..) of the therapeutic properties of mud under study.
There are also some comments according to definitions in the text.
Line 51. “ coastal mud deposits, which may contain elevated levels of minerals such as sulfur, selenium, iodine, and organic matter”. Sulfur, selenium, iodine, and organic matter are chemical elements but they are not minerals.
Line 65-66. “In addition, the physical and chemical properties of the sediments, such as grain size distribution, mineralogy, calcium carbonate, and organic content, play a significant role in the depositional processes of mud”. What means "mineralogy" and "calcium carbonate"? Calcium carbonate is a mineral – calcite. It is mineralogy.
Comments on the Quality of English LanguageMinor editing of English language required
Reviewer 3 Report
Comments and Suggestions for Authors
The topic of this article seems not very focused. The whole article mainly analyzes the sedimentary characteristics of the two islands. The analysis of the sedimentary environment does not seem to be well related to the role of therapeutic use. What is the scientific significance of this article? If the importance of the study is not well explained, it is just a local study and has no scientific significance.
The structure of the article is not very reasonable, and the fourth and fifth parts take up too much space. If these two parts were just an introduction to the geological background, they could be completely simplified.
Except for CaCO3, and TOC, geochemical analysis should also include some element analysis such as major, trace or REE elements, etc. As mentioned by the authors in the introduction part, some elements such as sulfur, selenium, iodine, which are key elements to the therapeutic values of sediment, should also analyzed in geochemical analysis.
In addition, pH, plasticity, surface characteristics, and thermal properties should be also analyzed and added in the physical-chemical properties of sediment, as well as grain size, which might have influences on the therapeutic use of the sediment. Grain size should not be included in the geochemical analysis. Authors mentioned that the paper will offer a comprehensive overview of the mineralogical, geochemical, and physical characteristics of the materials, however no any mineralogical analysis of the samples was seen in the manuscript.
Only part 7.6 seems to deal with relationship between the physical-chemical characteristics of sediment and therapeutic value. However, the conclusion is not convincing enough. The connection between sediment classification and therapeutic value should also be explained in depth. Especially, the role of sedimentary conditions of shallow marine and tidal waters for the therapeutic use needs to be discussed in detail.
Round 2
Reviewer 2 Report
Comments and Suggestions for Authors
Authors took into consideration all comments and improved the text in explanation of therapeutic application of mud. The paper can accept in present form.
Reviewer 3 Report
Comments and Suggestions for Authors
The quality of the manuscript have been improved after revision. I recommend its publication on the Geosciences.